# Zero-Forgetting Class-Incremental Segmentation via Dual-Phase Cognitive Cascades

## Abstract

Continual semantic segmentation (CSS) is a cornerstone task in computer vision that enables a large number of downstream applications, but faces the catastrophic forgetting challenge. In conventional class-incremental semantic segmentation (CISS) frameworks using Softmax-based classification heads, catastrophic forgetting originates from Catastrophic forgetting and task affiliation probability. We formulate these problems and provide a theoretical analysis to more deeply understand the limitations in existing CISS methods, particularly Strict Parameter Isolation (SPI). To address these challenges, we follow a dual-phase intuition from human annotators, and introduce **Cog**nitive **Ca**scade **S**egmentation (CogCaS), a novel dual-phase cascade formulation for CSS tasks in the CISS setting. By decoupling the task into class-existence detection and class-specific segmentation, CogCaS enables more effective continual learning, preserving previously learned knowledge while incorporating new classes. Using two benchmark datasets PASCAL VOC 2012 and ADE20K, we have shown significant improvements in a variety of challenging scenarios, particularly those with long sequence of incremental tasks, when compared to exsiting state-of-the-art methods. Our code will be made publicly available upon paper acceptance.

## 1 Introduction

Deep learning has transformed semantic segmentation into a cornerstone of computer vision, enabling influential applications from autonomous navigation to medical diagnostics. Despite these advances, real-world deployment faces a fundamental limitation, i.e., the conventional paradigm requires all object categories to be predefined before training. When new classes emerge, as they inevitably do in dynamic environments, models must be retrained with training data of both old and new classes, incurring increased computational costs and potential privacy concerns.

This limitation has driven substantial research in continual learning (CL), where models incrementally acquire new knowledge while trying to preserve existing capabilities. The core challenge is catastrophic forgetting: the tendency of neural networks to abruptly forget previously learned information when updated. Current approaches span multiple paradigms, including regularization-based methods Wang et al. (2022); Xu et al. (2021) which constrain parameter updates, replay-based strategies Aljundi et al. (2021); Rusu et al. (2022) which maintain historical exemplars, and optimization techniques Li et al. (2022) which seek non-interfering parameter spaces. Among these, SSUL Cha et al. (2021) stands out for providing theoretical guarantee of zero-forgetting through parameter compartmentalization, effectively addressing the stability-plasticity dilemma in classification tasks.

However, extending continual learning to semantic segmentation introduces unique complexities beyond classification. In particular in class-incremental semantic segmentation (CISS), models must adapt to new categories while maintaining pixel-precise understanding of previous classes, all without access to complete historical data. This setting introduces the challenge of background shift: pixels belonging to future classes are temporarily labeled as background, requiring the model to continuously revise its understanding of what constitutes "background" as new classes emerge Cermelli et al. (2020b). This continuous re-evaluation of the "background" class due to background shift critically exacerbates the stability-plasticity dilemma, often compelling the model to make a detrimental trade-off between the forgetting of previously learned classes and the insufficient acquisition of new ones, a challenge visually depicted in Figure 1.

While recently developed CISS approaches show promising performance in knowledge preservation, we prove that this localized optimization strategy, even assuming perfect preservation of knowledge of past classes, structurally prevents convergence to the global optimum achievable through joint training on all classes. Our theoretical analysis also reveals that they face fundamental limitations. Architecturally, the commonly used task headers based on Softmax need to output complete probabilities for all current class at every stage. However, the probability output for task segmentation head is essentially local and is only optimized for the current task category. This creates a stark trade-off: while freezing historical heads eliminates dynamic interference, it simultaneously cements distributional biases that prevent global optimization.

Motivated by these theoretical analyses, we propose a **Cog**nitive **Ca**scade **S**egmentation (CogCaS) architecture that fundamentally restructures the CISS paradigm. Our approach introduces two key innovations, i.e., **Existence-Driven Activation** by which segmentation heads are activated only for detected classes and thus background interference is eliminated, and **Parameter Modularity** by which independent detector and segmenter per class enable isolated evolution without cross-task contamination. Our implementation deliberately adopts elementary components: basic backbones, single-layer detectors and simple segmenters, and standard loss functions. This architectural transparency ensures that our empirical improvements stem purely from the cognitive cascade design rather than implementation sophistication, while maintaining the framework's extensibility for future enhancements.

In summary, our main contributions include (1) the first systematic characterization of the advantages and disadvantages of SPI strategy in CISS, (2) a cognitively inspired architecture that resolves the stability-plasticity dilemma through decoupling design, and (3) state-of-the-art performance with significantly increased margins particularly in challenging long-sequence continual learning scenarios.

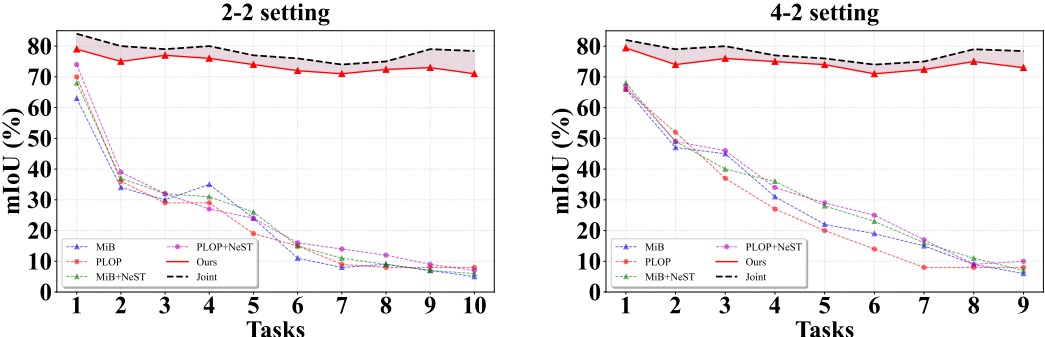

Figure 1: mIoU of classes learned at each task in PASCAL VOC continual semantic segmentation under 2-2 (left) and 4-2 (right) settings. While existing methods exhibit declining learning capability as tasks progress, our CogCaS consistently achieves strong performance around 70% mIoU, demonstrating stable and robust learning ability across all incremental stages.

## 2 RELATED WORK

Class-Incremental Semantic Segmentation (CISS) as a relatively new research topic is often considered as an extension of image-level continual learning, and therefore those techniques developed originally for image-level continual learning have been adopted for CISS, including *regularization* Cermelli et al. (2020a), *pseudo-labeling/self-supervision* Xie et al. (2024); Yang et al. (2023a), and *experience replay* Cha et al. (2021); Maracani et al. (2021a); Yang et al. (2023b); Zhang et al. (2022b). Such adoption is still being actively explored, as shown by very recent attempts Yu et al. (2025). All these methods model segmentation as *per-pixel multi-class Softmax classification*, where old and new logits directly compete, causing background drift that accumulate with the number of tasks Farajtabar et al. (2020) which can be seen in Figure 1.

To mitigate Softmax competition, several studies recast CISS as a *pixel-level multi-label* problem combined with strict parameter freezing. SSUL (Cha et al., 2021) appends binary channels for each

class and freezes them thereafter, but relies heavily on the initial task's class capacity to initialize its "unknown" class prediction; in dense incremental settings (e.g., 1-1), this causes performance collapse as incomparable logits from disjoint phases are forced into global competition. IPSeg (Yu et al., 2025) extends this paradigm with image posterior calibration but still requires exemplar replay for stability. In addition, built on Mask2Former (Cheng et al., 2021), methods such as CoMFormer (Cermelli et al., 2023) and CoMaSTRe (Gong et al., 2024) treat CISS as a *mask-classification* problem. CoMaSTRe decouples "where" and "what" via query-based segmenters and objectness transfer, yet relies on complex multi-stage distillation to mitigate forgetting. ECLIPSE (Kim et al., 2024) employs visual prompt tuning within a shared decoder and logit manipulation to address drift in continual panoptic segmentation. **Our *CogCaS* adopts *existence-driven activation*, which decouples "does class $c$ exist?" from "segment class $c$", and freezes each class-specific segmentation head after learning, achieving task-agnostic zero forgetting while maintaining high plasticity for new classes.** Crucially, CogCaS is derived from Hessian-based theoretical analysis (section 3.2): the explicit image-level router activates independent binary heads, structurally eliminating cross-task competition without distillation or replay.

## 3 METHODOLOGY

### 3.1 PRELIMINARIES

**Notations.** We consider the problem of Class-Incremental Semantic Segmentation (CISS), where a model sequentially learns a sequence of $T$ tasks, denoted as $\{\mathcal{T}_1, \ldots, \mathcal{T}_T\}$. Each task $\mathcal{T}_t$ is associated with a unique training set $\mathcal{D}_t$ which contains certain number of training images and corresponding pixel-wise annotations. The task introduces a set of task-specific foreground classes $\mathcal{C}_t$, which are mutually exclusive between tasks, i.e., $\mathcal{C}_\tau \cap \mathcal{C}_t = \emptyset$ for all $\tau \neq t$. For each training image in task $\mathcal{T}_t$, each pixel is annotated as one of the task-specific foreground classes in $\mathcal{C}_t$ or the special "background" class. Notably for the training set $\mathcal{D}_t$, all pixels not belonging to the foreground class set $\mathcal{C}_t$ are annotated as background. Thus, the annotated background regions in each training image of task $\mathcal{T}_t$ encompass the true background as well as image regions corresponding to foreground classes from past ($\mathcal{T}_{1:t-1}$) or future ($\mathcal{T}_{t+1:T}$) tasks. When updating the model with the training set $\mathcal{D}_t$ and validation set $\mathcal{D}_t^v$ of task $\mathcal{T}_t$, the model needs to be expanded to account for the set $\mathcal{C}_t$ of new foreground classes. Let $\boldsymbol{\theta}$ denote the collection of all the learnable model parameters throughout the continual learning process (from task $\mathcal{T}_1$ to $\mathcal{T}_T$). When the model learns task $\mathcal{T}_t$, the part of $\boldsymbol{\theta}$ which are uniquely associated with future tasks ($\mathcal{T}_{t+1:T}$) are frozen with certain default value (i.e., zero here). After updating the model based on certain task-specific loss function $\mathcal{L}_t$, the leranable model parameters are changed from $\boldsymbol{\theta}_{t-1}^*$ to $\boldsymbol{\theta}_t^*$, where $\boldsymbol{\theta}_t^*$ represents the locally optimal model parameters that minimizes $\mathcal{L}_t$. The change in parameter values from $\mathcal{T}_{t-1}$ to $\mathcal{T}_t$ is denoted by $\Delta_t := \boldsymbol{\theta}_t^* - \boldsymbol{\theta}_{t-1}^*$, and the Hessian matrix of the loss function $\mathcal{L}_t$ with respect to the learnable parameters $\boldsymbol{\theta}$ is denoted by $\mathbf{H}_t(\boldsymbol{\theta}) = \frac{\partial^2 \mathcal{L}_t(\boldsymbol{\theta})}{\partial \boldsymbol{\theta}^2}$.

**Convergence Assumption.** To enable formal analysis of the CISS framework, we make the following assumption about the learning dynamics.

**Assumption 3.1** (Convergence for each task). For each task $\mathcal{T}_t$, the optimization process converges to a locally optimal model parameters $\boldsymbol{\theta}_t^*$, such that within its neighborhood $\mathcal{N}(\boldsymbol{\theta}_t^*)$, the magnitude of the loss gradient $\nabla \mathcal{L}_t(\boldsymbol{\theta})$ is bounded (smaller than $\epsilon$) and the Hessian matrix $\mathbf{H}_t(\boldsymbol{\theta})$ remains positive semi-definite, satisfying the second-order conditions for local optimality, i.e.,

$$|\nabla \mathcal{L}_t(\boldsymbol{\theta})| \le \epsilon, \mathbf{H}_t(\boldsymbol{\theta}) \succeq 0, \quad \forall \boldsymbol{\theta} \in \mathcal{N}(\boldsymbol{\theta}_t^*). \tag{1}$$

Furthermore, it assumes that the magnitude of parameter updates satisfies $|\Delta_t| < \delta$, $\forall t \le T$, for some small constant $\delta > 0$. When learning a new task (per Assumption 3.1), the model's performance on prior tasks can degrade, a phenomenon known as catastrophic forgetting. To quantify this, we will define the forgetting rate (see below) based on the change in the loss function, which serves as a continuous and differentiable measurement function for task performance and reflects generalization performance when evaluated on a validation set.

**Definition 3.2** (Average Forgetting Rate). The forgetting rate for a previously learned task $\mathcal{T}_\tau$ (where $\tau < t$) after the model parameters have been updated to $\theta_t$ for task $\mathcal{T}_t$, is defined as the change in the loss evaluated on the validation set $\mathcal{D}_\tau^v$. Formally, it is given by $\mathcal{E}_\tau(\theta_t) = \mathcal{L}_\tau^{val}(\theta_t) - \mathcal{L}_\tau^{val}(\theta_\tau^*)$,

where $\mathcal{L}_\tau^{val}$ denotes the loss computed on the validation data for task $\mathcal{T}_\tau$. By definition, this rate is zero when evaluated at the task's own optimal parameters, i.e., $\mathcal{E}_\tau(\theta_\tau^*) = 0$. In the context of continual learning, the locally optimal parameters $\theta_\tau^*$ for task $\mathcal{T}_\tau$ may cause forgetting of knowledge from previous tasks $\mathcal{T}_{1:\tau-1}$. This effect is measured by the Average Forgetting Rate, defined as

$$\bar{\mathcal{E}}_t(\theta_t^*) = \frac{1}{t-1}\sum_{\tau=1}^{t-1}\mathcal{E}_\tau(\theta_t^*) \quad t \geq 2 . \tag{2}$$

## 3.2 Strict Parameter Isolation in CISS

Based on Assumption 3.1 and Definition 3.2, the relationship between the average forgetting rate for task $\mathcal{T}_t$ and $\mathcal{T}_{t-1}$ can be obtained as (see Appendix B.1)

$$\bar{\mathcal{E}}_t(\theta_t^*) = \frac{1}{t-1}\left((t-2)\cdot\bar{\mathcal{E}}_{t-1}(\theta_{t-1}^*) + \frac{1}{2}\Delta_t^\intercal(\sum_{\tau=1}^{t-1}\mathrm{H}_\tau(\theta_\tau^*))\Delta_t + v^\intercal\Delta_t\right) + \mathcal{O}(\delta\cdot\epsilon) , \tag{3}$$

where $v^T = \sum_{\tau=1}^{t-1}(\theta_{t-1}^* - \theta_\tau^*)^\intercal\mathbf{H}_\tau(\theta_\tau^*)$. From Equation (3), it is clear that the average forgetting rate $\bar{\mathcal{E}}_t(\theta_t^*)$ of locally optimal parameters $\theta_t^*$ for task $\mathcal{T}_t$ is directly related to the average forgetting rate $\bar{\mathcal{E}}_{t-1}(\theta_{t-1}^*)$ of locally optimal parameters $\theta_{t-1}^*$ for task $\mathcal{T}_{t-1}$. With such relationship, we can obtain the following zero-forgetting condition (see Appendix B.1).

**Theorem 3.3** (Zero-forgetting Condition). *For any continuous learning algorithm that satisfies Assumption 3.1, (1) if $\bar{\mathcal{E}}_\tau(\theta_\tau^*) = 0$, $\forall\tau < t$, then $\bar{\mathcal{E}}_t(\theta_t^*) = \frac{1}{2(t-1)}\Delta_t^\intercal\left(\sum_{i=1}^{t-1}\mathbf{H}_i(\theta_i^*)\right)\Delta_t$, and (2) $\mathcal{E}_\tau(\theta_t^*) = 0, \forall\tau < t$, if and only if $\Delta_t^\intercal(\sum_{\tau=1}^{t-1}\mathbf{H}_\tau(\theta_\tau^*))\Delta_t = 0$.*

Theorem 3.3 offers two significant implications for achieving zero forgetting. First, even if zero average forgetting ($\bar{\mathcal{E}}_\tau(\theta_\tau^*) = 0$) is achieved for all previous tasks $\tau < t$ when evaluated at their respective optimal parameters, learning a new task $\mathcal{T}_t$ and thereby updating parameters (resulting in $\Delta_t \neq 0$) can still cause considerable average forgetting. Second, Theorem 3.3 (second half) provides a direct mathematical condition for achieving true zero forgetting on all past tasks (i.e., $\mathcal{E}_\tau(\theta_t^*) = 0, \forall\tau < t$) after the model learns task $\mathcal{T}_t$, i.e., the quadratic term $\Delta_t^\intercal\left(\sum_{\tau=1}^{t-1}\mathbf{H}_\tau(\theta_\tau^*)\right)\Delta_t$ must be zero. Consequently, any continual learning algorithm aiming for zero forgetting must be designed to ensure this quadratic term vanishes. Indeed, existing zero-forgetting methods, such as orthogonal gradient method Farajtabar et al. (2020) and projected gradient method Saha et al. (2021), work by satisfying such a condition (see details in Appendix B.2).

Strict parameter isolation (SPI) which is used in Cha et al. (2021); Yu et al. (2025) is a strategy that can satisfy the zero forgetting condition in Theorem 3.3. By optimizing a unique set of parameters for each new task while freezing those for previous tasks, the parameter update $\Delta_t$ is guaranteed to be in a subspace orthogonal to the parameters of all previous tasks. Consequently, the quadratic term $\Delta_t^\intercal(\sum_{\tau=1}^{t-1}\mathbf{H}_\tau)\Delta_t$ is always zero, thus ensuring theoretical zero-forgetting.

Although the SPI strategy can theoretically prevent catastrophic forgetting by isolating task-specific parameters, its direct application in CISS introduces a critical challenge: the problem of incomparable outputs. In the SPI framework, each task-specific segmentation head is trained independently on a subset of classes. Consequently, the output logits from different heads are not mutually calibrated; a high score from one head is not directly comparable to a score from another trained on a different task. This renders the standard approach of applying a simple $argmax$ operation across all heads' outputs to determine the final class for each pixel fundamentally flawed (also can be seen in Figure 2(A)). This issue stems from the SPI model's inability to determine a pixel's task affiliation before classifying it. This problem is conceptually analogous to challenges in image-level continual learning. A prior study Kim et al. (2022), for instance, factorizes the image-level prediction into an intra-task prediction and a task affiliation probability. Borrowing this formulation for our pixel-level problem, the prediction for a pixel $y_p$ can be written as:

$$P\left(y_p = c \mid \mathbf{x}\right) = P\left(y_p = c \mid \mathbf{x}, \mathcal{T}_t\right)\cdot P\left(\mathcal{T}_t \mid \mathbf{x}\right) , \quad \forall c \in \mathcal{C}_t , \tag{4}$$

where $p$ is the index of a pixel in the input image $\mathbf{x}$, and $c$ is a class learned from task $\mathcal{T}_t$. While SPI perfectly preserves the intra-task prediction term $P(y_p = c|x, \mathcal{T}_t)$ due to its zero-forgetting nature. There is no mechanism to estimate the crucial task affiliation probability $P(\mathcal{T}_t|x)$, thus hindering effective final prediction.

Figure 2: Demonstration of existing typical CISS framework and the proposed CogCaS framework. (A) Traditional CISS framework consists of a set of segmentation heads which are trained sequentially and activated for inference simultaneously. (B) our proposed CogCaS restructures the traditional CISS formulation into a dual-phase cascade using multi-label classifier and class-specific segmentation head. The multi-label classifier determine whether each learned class exists in the image, and only segmentation heads corresponding to existing classes are activated to produce a foreground-background mask. These masks are then fused to obtain the final segmentation mask using a mask fusion strategy.

### 3.3 CLASS-INCREMENTAL SEGMENTATION VIA COGNITIVE CASCADES

To resolve the issue of incomparable outputs in Equation (4) and fully leverage the zero-forgetting property of SPI, we introduce **Cog**nitive **Cas**cade Segmentation (CogCaS), a novel framework that fundamentally restructures the CISS paradigm. Our approach is motivated by the coarse-to-fine strategy employed by human annotators who first identify the object categories present in an image before delineating their precise boundaries. CogCaS mimics this cognitive process by decoupling the problem into two sequential phases. First, an image-level classifier which acts as a Task Router determines the existence of all learned classes within the input image. Second, only the parameter-isolated segmentation heads corresponding to the detected classes are activated to perform binary foreground-background segmentation. This restructuring can be formalized(in Appendix B.4) by reframing the probabilistic decomposition from Equation (4) into a more intuitive, class-centric model:

**Notation Clarification.** Let $Y_p \in \mathcal{C}_{[1:t]} \cup \{0\}$ denote the random variable representing the class label of pixel $p$, and let $Z_c \in \{0,1\}$ be a binary random variable indicating whether class $c$ is present in image $\mathbf{x}$ (i.e., $Z_c = 1$ if $\exists p : Y_p = c$). The cascade decomposition is then:

$$P(Y_p = c \mid \mathbf{x}) \propto \underbrace{P\big(Y_p = c \mid \mathbf{x}, Z_c = 1\big)}_{\text{Binary segmentation}} \cdot \underbrace{P\big(Z_c = 1 \mid \mathbf{x}\big)}_{\text{Class existence (router)}}, \quad \forall c \in \mathcal{C}_{[1:t]}, \tag{5}$$

where the first term is the conditional probability that pixel $p$ belongs to class $c$ given that class $c$ exists in the image, and the second term is the existence probability estimated by the multi-label classifier. In practice, we threshold the existence probability $P(Z_c = 1|\mathbf{x}) \geq \alpha$ to obtain the predicted class set $\mathcal{C}_{pred} = \{c : P(Z_c = 1|\mathbf{x}) \geq \alpha\}$.

The overall pipeline of this framework is depicted in Figure 2(B). An input image $\mathbf{x}$ is first passed through a task-shared, frozen pretrained feature extractor $\Phi$ to generate feature maps $F = \Phi(\mathbf{x})$. Subsequently, $F$ is processed by the following two cascade phases.

**Phase I: Image-Level Category Recognition.** This phase is designed to infer the presence of each learned class. We use a multi-label classifier which operates on the feature maps $F$ to yield a class-existence probability $P(Z_c = 1|\mathbf{x})$ for each learned class. The set of predicted classes

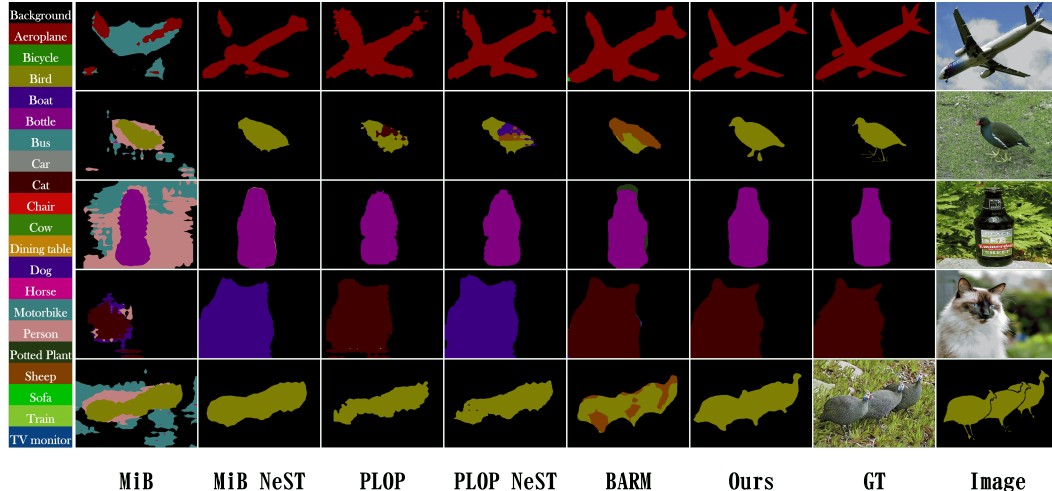

Figure 3: Representative segmentation results from different methods after the model learns all tasks in the Pascal VOC 2012 15-1 setting.

$C_{pred}$ is then identified by applying a threshold $\alpha$ to these probabilities, i.e., $\mathcal{C}_{pred} = \{c : P(Z_c = 1|\mathbf{x}) \geq \alpha\}$. Specifically, we adopt a Sigmoid function with a Binary Cross-Entropy loss, which is naturally suited for the multi-label classification setting. To enable continual learning, each class-specific weight block $\theta_c^{cls}$ is frozen after its corresponding training task is complete. This design not only achieves Strict Parameter Isolation (SPI) to circumvent catastrophic forgetting, but also fundamentally avoids the problem of cross-task output comparison by directly learning independent class-existence probabilities.

**Phase II: Class-Specific Binary Segmentation.** This phase is responsible for estimating the conditional segmentation term, $P(Y_p = c|\mathbf{x}, Z_c = 1)$. For each class $c \in C_{pred}$ identified in Phase I, its dedicated and parameter-isolated segmentation head $H_c(\cdot)$ is activated. This head takes the feature maps $F$ as input and produces a two-channel probability map, $M_c = (m_c^{bg}, m_c^{fg})$, representing the probability of each pixel belonging to the "relative background for class c" and the "foreground for class c," respectively. The binary segmentation schema offers key advantages: it simplifies the complex multi-class decision boundary into a single binary decision boundary, leading to faster convergence and higher fidelity.

**Mask Fusion.** Since Phase II yields independent binary masks, a fusion step $\mathcal{U}$ is required to integrate them into a coherent multi-class segmentation map by resolving overlapping predictions. We evaluate several fusion strategies and analyze their impact in table 6.

**Precise Model Lifecycle.** To ensure reproducibility and clarity, we provide a rigorous description of the complete model lifecycle:

*Instantiation.* Given an input image $\mathbf{x}$, features are extracted as $F = \Phi(\mathbf{x})$, where $\Phi$ is a frozen pretrained backbone (e.g., ResNet-101 or Swin-L). When learning a new task $\mathcal{T}_t$, we instantiate *only* new parameters: (i) router weights $\theta_c^{cls}$ for each new class $c \in \mathcal{C}_t$, and (ii) binary segmentation heads $H_c$ with parameters $\theta_c^{seg}$ for each $c \in \mathcal{C}_t$. All parameters from previous tasks $\mathcal{C}_{1:t-1}$ remain strictly frozen.

*Training.* For task $\mathcal{T}_t$, we update *only* the newly instantiated parameters $\{\theta_c^{cls}, \theta_c^{seg}\}_{c \in \mathcal{C}_t}$. Importantly, the multi-label classifier (Phase I) and segmentation heads (Phase II) are trained *separately* to avoid interference. The training losses for each component are detailed in section B.5. Gradients are computed only for $\{\theta_c^{cls}, \theta_c^{seg}\}_{c \in \mathcal{C}_t}$; all other parameters have zero gradient by design (SPI).

*Inference.* At test time, we first compute existence probabilities for all learned classes, then activate only the segmenters corresponding to detected classes, and finally fuse their binary masks into the final segmentation map. The complete inference pipeline is detailed in section B.5.

## 4 EXPERIMENTS

### 4.1 SETUP

**Datasets.** Following prior work Cermelli et al. (2020a); Cha et al. (2021); Maracani et al. (2021b); Yang et al. (2023a), the proposed CogCaS was evaluated on two semantic segmentation datasets PASCAL VOC Everingham et al. (2010) and ADE20K Zhou et al. (2017) with different complexity levels. PASCAL VOC contains 20 object classes plus a background class, with 10,582 samples for training and 1,449 samples for validation, while the large-scale dataset ADE20K presents a more challenging scenario, containing 150 foreground classes and one background class, with 20,210 and 2,000 samples for training and validation, respectively.

**CISS Settings.** With each dataset, the widely used $M$-$N$ setting is adopted, with $M$ being the number of foreground classes in the first task and $N$ the number of new foreground classes in each subsequent task. For example, in the VOC 10-1 setting, the model first learns to segment 10 classes, then incrementally learns one new class in each subsequent task.

**Implementation details.** The proposed CogCaS was trained using 8 NVIDIA GeForce RTX 4090 GPUs. We conducted training on models utilizing both ResNet-101 He et al. (2016) and Swin-L Liu et al. (2021) backbones. The Adam optimizer Kingma & Ba (2015) was employed for training, with each task being trained for 90 epochs. The initial learning rate was set to $1 \times 10^{-5}$. More details can be found in the Appendix B.5.

### 4.2 MAIN RESULTS

Experimental results demonstrate the efficacy of the proposed CogCaS method on the PASCAL VOC 2012 and ADE20K datasets, as presented in Table 1 and Table 2 respectively.

Our CogCaS as a non-replay method was compared with basic and state-of-the-art non-replay baselines. As Table 1 shows, on the PASCAL VOC 2012 dataset, CogCaS exhibited superior performance across various incremental learning configurations. For example, in the VOC 10-1 setting (11 tasks), CogCaS achieved the highest mean Intersection over Union (mIoU) of 70.2% for new classes (11-20) and a leading overall mIoU of 72.1%. The superiority of CogCaS was more pronounced on the complex ADE20K dataset across all evaluated incremental settings (Table 2). Figure 3 visually confirms the superior performance of our method. These results consistently support the efficacy of our CogCaS in learning new knowledge and preserving old knowledge in both small-scale and large-scale incremental scenarios.

We also compare with recent large-model-based CISS methods (e.g., SAM-based DecoupleCSS with ∼632M parameters) in table 7. Our architecture is orthogonal to backbone choice and offers a favorable balance between performance and efficiency.

To further confirm the robustness of our method, experiments under more challenging conditions were performed in which tasks are more numerous with fewer classes to be learned within each class. As Table 3 shows, our CogCaS significantly outperforms traditional knowledge distillation methods (MiB, PLOP, and NeST variants) and parameter-isolation strategies (SSUL, IPSeg). For example, in the VOC 1-1 setting (totally 20 tasks), our method achieves 70.9% overall mIoU versus only 7.3% for PLOP+NeST and 33.2% for IPSeg. In the VOC 2-2 setting (10 tasks), our method reaches 71.5% compared to 27.8% (MiB+NeST) and 51.4% (IPSeg). To verify statistical robustness, we re-evaluated the VOC 1-1 setting using three independent random seeds, yielding an average mIoU of 71.1% (± 0.8%), which is highly consistent with the reported 70.9%. This confirms that the substantial performance gap over IPSeg (33.2%) is statistically stable. The low variance stems from the inherent stability of our SPI design: increasing task count only adds training cost without affecting previously learned knowledge.

These results clearly demonstrate our CogCaS can well learn new classes and preserve old knowledge even in a long CISS learning process.

To make the first phase explicit, Table 4 summarises class-existence detection metrics (mAP, precision, and recall) on both benchmarks.

Table 4: Phase I class-existence detection on the evaluation split (%). Results are averaged across all settings in the datasets.

| Dataset | mAP↑ | Prec.↑ | Rec.↑ |
|---|---|---|---|
| PASCAL VOC 2012 | 85.12 | 93.47 | 90.25 |
| ADE20K | 79.84 | 73.86 | 78.03 |

Table 1: Comparison with exsiting CISS methods on PASCAL VOC 2012 in mIoU (%). The best results are marked in **bold**. ○: ResNet101 backbone. ◇: Swin-L backbone. †: unlike other methods, this one is based on Mask2Former Cheng et al. (2021).

| Method | **10-1** (11 tasks) | | | **15-5** (2 tasks) | | | **15-1** (6 tasks) | | |
|---|---|---|---|---|---|---|---|---|---|
| | 0-10 | 11-20 | all | 0-15 | 16-20 | all | 0-15 | 16-20 | all |
| **Ours○** | **73.9** | **70.2** | **72.1** | 75.5 | **70.3** | 74.1 | 75.5 | **71.4** | **74.4** |
| **Joint Deeplab-v3○** Chen et al. (2017) | 78.4 | 76.4 | 77.4 | 79.8 | 72.4 | 77.4 | 79.8 | 72.4 | 77.4 |
| **Joint Ours○** | 79.3 | 77.6 | 78.4 | 79.2 | 76.2 | 78.4 | 79.2 | 76.2 | 78.4 |
| LwF-MC○ Rebuffi et al. (2017) | 4.7 | 5.9 | 5.0 | 58.1 | 35.0 | 52.3 | 64.0 | 8.4 | 6.9 |
| ILT○ Michieli & Zanuttigh (2021) | 7.2 | 3.7 | 5.5 | 67.1 | 39.2 | 60.5 | 8.8 | 8.0 | 8.6 |
| MiB○ Cermelli et al. (2020a) | 31.5 | 13.1 | 22.7 | 71.8 | 43.3 | 64.7 | 46.2 | 22.9 | 40.7 |
| MiB+NeST○ Xie et al. (2024) | 39.4 | 21.1 | 30.6 | 75.5 | 48.7 | 69.5 | 60.2 | 29.9 | 53.0 |
| PLOP ○ Douillard et al. (2021) | 44.0 | 15.5 | 30.5 | 75.4 | 49.6 | 69.3 | 64.1 | 20.1 | 53.1 |
| PLOP+NeST○ Xie et al. (2024) | 47.2 | 16.3 | 32.4 | 77.6 | 55.8 | 72.4 | 67.2 | 25.7 | 57.3 |
| BARM○ Zhang & Gao (2024) | 72.2 | 49.8 | 61.9 | 74.9 | 69.5 | 73.6 | **77.3** | 45.8 | 61.9 |
| PLOP+LCKD○ Yang et al. (2023a) | — | — | — | 75.2 | 54.8 | 71.1 | 69.3 | 30.9 | 61.1 |
| SSUL○ Cha et al. (2021) | 71.3 | 46.0 | 59.3 | 77.8 | 50.1 | 71.2 | **77.3** | 36.6 | 67.6 |
| RCIL○ Zhang et al. (2022a) | 55.4 | 15.1 | 34.3 | **78.8** | 52.0 | 72.4 | 70.6 | 23.7 | 59.4 |
| IDEC○ Zhao et al. (2023) | 70.7 | 46.3 | 59.1 | 78.0 | 51.8 | 71.8 | 77.0 | 36.5 | 67.3 |
| **Ours◇** | **76.1** | **75.7** | **75.9** | 78.3 | **74.9** | **77.8** | 78.4 | **72.5** | **76.9** |
| **Joint Deeplab-v3◇** Chen et al. (2017) | 81.4 | 78.4 | 79.9 | 80.8 | 77.3 | 79.9 | 80.8 | 77.3 | 79.9 |
| **Joint Ours◇** | 82.7 | 80.9 | 81.8 | 81.3 | 83.4 | 81.8 | 81.3 | 83.4 | 81.8 |
| MicroSeg◇ Zhang et al. (2022b) | 73.5 | 53.0 | 63.8 | **81.9** | 54.0 | 75.2 | **80.5** | 40.8 | 71.0 |
| MiB◇ Cermelli et al. (2020a) | 35.7 | 14.8 | 26.7 | 74.3 | 45.1 | 67.3 | 48.7 | 19.5 | 41.7 |
| MiB+NeST◇ Xie et al. (2024) | 41.3 | 24.1 | 33.1 | 77.8 | 50.1 | 71.2 | 63.2 | 23.5 | 53.7 |
| PLOP◇ Douillard et al. (2021) | 47.2 | 18.4 | 33.5 | 79.2 | 50.2 | 72.3 | 67.6 | 25.2 | 57.6 |
| PLOP+NeST○ Xie et al. (2024) | 49.2 | 19.8 | 35.2 | 81.6 | 55.8 | 75.4 | 72.2 | 33.7 | 63.1 |
| BARM○ Zhang & Gao (2024) | 74.2 | 53.8 | 64.4 | 77.8 | 72.1 | 76.4 | 79.3 | 48.1 | 71.8 |
| SSUL◇ Cha et al. (2021) | 74.3 | 51.0 | 63.2 | 79.7 | 55.3 | 73.9 | 78.1 | 33.4 | 67.5 |
| CoMasTRe◇† Gong et al. (2024) | — | — | — | 79.7 | 51.9 | 73.1 | 69.8 | 43.6 | 63.5 |
| CoMFormer◇† Cermelli et al. (2022) | — | — | — | 74.7 | 54.3 | 71.1 | 70.8 | 32.2 | 61.6 |

Table 2: Comparison with existing CISS methods on ADE20K using Swin-L backbone. †: unlike other methods, this one is based on Mask2Former Cheng et al. (2021).

| Method | **100-50** (2 tasks) | | | **100-10** (6 tasks) | | | **100-5** (11 tasks) | | |
|---|---|---|---|---|---|---|---|---|---|
| | *0-100* | *101-150* | *all* | *0-100* | *101-150* | *all* | *0-100* | *101-150* | *all* |
| **Ours** | 41.2 | **29.4** | 37.3 | **42.3** | **25.6** | **36.8** | 40.1 | **24.7** | **35.0** |
| **Joint Deeplab-v3** Chen et al. (2017) | 47.2 | 31.8 | 42.1 | 47.2 | 21.8 | 42.1 | 47.2 | 21.8 | 42.1 |
| **Joint Ours** | 47.8 | 38.7 | 44.7 | 47.8 | 38.7 | 44.7 | 47.8 | 38.7 | 44.7 |
| MiB Cermelli et al. (2020a) | 39.0 | 16.7 | 31.2 | 36.6 | 9.8 | 27.7 | 34.7 | 4.8 | 24.7 |
| MiB+NeST Xie et al. (2024) | 38.8 | 23.1 | 33.5 | 38.8 | 19.1 | 32.2 | 35.2 | 13.6 | 28.1 |
| PLOP Douillard et al. (2021) | 40.4 | 13.4 | 31.5 | 39.4 | 12.6 | 30.1 | 36.9 | 6.2 | 26.7 |
| PLOP+NeST Xie et al. (2024) | 40.8 | 22.8 | 34.8 | 39.4 | 20.5 | 33.2 | 38.3 | 15.4 | 30.7 |
| BARM Zhang & Gao (2024) | 42.0 | 23.0 | 35.7 | 41.1 | 23.1 | 35.2 | 40.5 | 21.2 | 34.1 |
| FALCON Truong et al. (2025) | **45.9** | 29.1 | **40.3** | 41.1 | 23.2 | 35.2 | **40.8** | 18.9 | 33.5 |
| CoMFormer† Cermelli et al. (2023) | 44.7 | 26.2 | 38.4 | 40.6 | 15.6 | 32.3 | 39.5 | 13.6 | 30.9 |
| CoMasTRe† Gong et al. (2024) | 45.7 | 26.0 | 39.2 | 42.3 | 18.4 | 34.4 | **40.8** | 15.8 | 32.6 |

Table 3: Comparison with existing CISS methods under more challenging continual learning settings.

| Method | **VOC 1-1** (20 tasks) | | | **VOC 2-1** (19 tasks) | | | **VOC 2-2** (10 tasks) | | |
|---|---|---|---|---|---|---|---|---|---|
| | 0-1 | 2-20 | all | 0-2 | 3-20 | all | 0-2 | 3-20 | all |
| Ours | **79.4** | **70.1** | **70.9** | **76.3** | **71.2** | **71.9** | **75.9** | **70.8** | **71.5** |
| MiB Cermelli et al. (2020a) | 27.3 | 6.4 | 8.3 | 23.6 | 7.9 | 10.14 | 41.1 | 23.4 | 25.9 |
| PLOP Douillard et al. (2021) | 25.4 | 4.2 | 6.2 | 19.4 | 6.2 | 8.1 | 39.7 | 22.8 | 25.2 |
| MiB+NeST Xie et al. (2024) | 28.1 | 6.8 | 8.7 | 24.5 | 8.1 | 10.4 | 40.4 | 25.8 | 27.8 |
| PLOP+NeST Xie et al. (2024) | 32.5 | 4.6 | 7.3 | 20.1 | 7.9 | 10.5 | 38.1 | 23.5 | 25.5 |
| SSUL Cha et al. (2021) | 60.1 | 29.6 | 32.5 | 59.6 | 34.7 | 38.2 | 60.3 | 40.6 | 44.0 |
| IPSeg Yu et al. (2025) | 61.8 | 30.2 | 33.2 | 60.1 | 32.6 | 36.2 | 64.7 | 49.5 | 51.4 |

Table 5: Ablation experiments with respect to the classification head using parameters trained under different task settings

| Method | VOC 1-1 (20 tasks) | | | VOC 2-1 (19 tasks) | | | VOC 2-2 (10 tasks) | | | ADE 100-5 (11 tasks) | | |
|---|---|---|---|---|---|---|---|---|---|---|---|---|
| | 0-1 | 2-20 | all | 0-2 | 3-20 | all | 0-2 | 3-20 | all | 0-100 | 101-150 | all |
| Segmentation Only | 18.2 | 13.1 | 13.5 | 14.3 | 17.2 | 16.7 | 13.8 | 16.5 | 16.1 | 6.7 | 9.5 | 7.6 |
| Full Model | 79.4 | 70.1 | 70.9 | 76.3 | 71.2 | 71.9 | 75.9 | 70.8 | 71.5 | 40.1 | 24.7 | 35.0 |
| Oracle | 80.2 | 70.3 | 71.2 | 77.4 | 71.5 | 72.3 | 77.4 | 71.8 | 72.4 | 48.6 | 31.8 | 43.0 |

## 4.3 ABLATION STUDIES

To assess the practical impact of the classification head during inference, we conducted comparative experiments with three distinct model configurations. The first, termed the "Full Model", utilizes the complete model architecture. The second, the "Segmentation Only" version, deactivates the classification head during testing, relying solely on the segmentation heads learned during training. The third, the "Oracle" version, substitutes the classification head's output with ground truth labels to isolate the component's error contribution. By comparing these three settings using standard semantic segmentation metrics, we can precisely determine the classification head's actual contribution and significance to our decoupled segmentation framework at test time.

The ablation experiments of this study in Table 5 show that there is just small difference in performance between the "Oracle" configuration (using real labels) and the "complete model" (for example, both are 70.9% and 71.2%), indicating that the multi-label classifier of the model can effectively handle the 20 categories of VOCs. However, the performance of the "segmentation only" configuration declined, indicating that learning the relationship between foreground and background remains a challenge even with the inclusion of near-OOD data during training.

On the ADE20K dataset with more complex categories (150 classes), the mIoU configured with "Oracle" (43.0%) was significantly better than that of the "Full Model" (35.0%), revealing that the classifier encounters challenges when facing a large number of categories, and its errors have a significant impact on the segmentation performance. These results jointly prove that the classification head is a key component in this decoupling framework. Especially when there are many categories, its accuracy is crucial to the final segmentation effect, and removing the classification head usually leads to performance loss.

## 4.4 SENSITIVITY STUDY

**Mask Fusion Strategy Analysis.** To handle overlapping predictions in segmentation masks, we evaluate five fusion strategies: (1) *Logits-based*: selects the class with highest confidence; (2) *Random*: randomly chooses among overlapping predictions; (3) *Strict*: assigns overlapping pixels to background; (4) *Distributed*: prioritizes rare categories to preserve small objects, where rarity is determined by pixel-count statistics collected during training and overlapping pixels are assigned to the class with fewer total training pixels; and (5) *Loose*: accepts predictions containing the ground truth category. As shown in table 6, the *Loose* strategy achieves superior performance across all settings, followed by our *Distributed* approach. The *Logits-based* and *Random* strategies show comparable results, while *Strict* performs worst due to its conservative background assignment. These results demonstrate the importance of appropriate overlap handling in incremental segmentation.

Table 6: Sensitivity analysis of mask fusion strategies on VOC across three incremental settings. Results are reported in mIoU (%).

| Fusion Strategy | VOC 10-1 | VOC 15-5 | VOC 15-1 |
|---|---|---|---|
| Logits-based | 71.4 | 73.3 | 73.4 |
| Random | 71.9 | 73.2 | 73.9 |
| Strict | 70.9 | 72.8 | 73.0 |
| Distributed (Ours) | 72.1 | 74.1 | 74.4 |
| **Loose** | **72.8** | **74.9** | **75.2** |

## 4.5 ADDITIONAL STUDY

Due to limited resources and time, our further investigations focused on the challenging 2-2 settings on the VOC dataset. To further validate our method, we unfroze the Encoder's final bottleneck layer, making it task-shared and trainable, as shown in Figure 4. Manually setting these task-shared parameters did not affect new class learning (matching baseline performance). However, this con-

figuration, when combined with SPI, resulted in catastrophic forgetting, evidenced by a 37% drop in mIoU. By intentionally disrupting the SPI settings, we observed that while new classes learned normally, old classes experienced catastrophic forgetting. This observation, in reverse, further substantiates the correctness of our proposed method. What's more, adding class-specific LoRA to the backbone, the improvement obtained by our method is not significant (only improve 1% mIou), but the parameters need to be changed with the task, which will significantly increase the inference time.

## 5 CONCLUSION

This study provides a theoretical analysis to deeply understand the limitations within existing CISS methods and introduces a novel dual-phase CISS framework in which the segmentation task is decomposed into two disentangled stages. Crucially, the SPI strategy inherent in our design enables the framework to **achieve a zero-forgetting rate** for knowledge learned in previous tasks. Consequently, the model's performance is not significantly affected by the number of tasks in the CISS setting, showing notable robustness in challenging, long-sequence scenarios where other methods falter. A primary limitation is that although our model's performance can approach the upper bound set by joint training, it demands substantial training resource. Future work includes

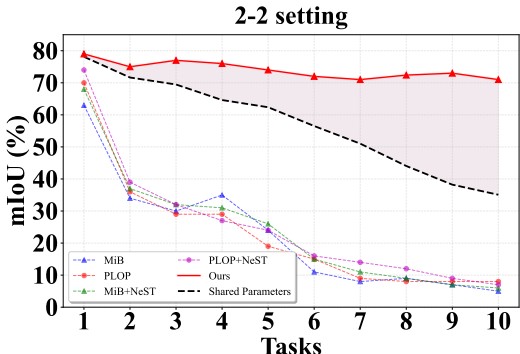

Figure 4: This figure shows the impact on the model performance when the task-shared parameters are set.

the investigation of class-specific fine-tuning of the feature encoder for other imaging modalities (e.g., medical images) and applying existing foundational models such as Segment Anything Model (SAM) as the segmentation heads in the proposed framework.

## ETHICS STATEMENT

This work adheres to the ICLR Code of Ethics. In this study, no human subjects or animal experimentation was involved. All datasets used, including ADE20K and PASCAL VOC 2012, were sourced in compliance with relevant usage guidelines, ensuring no violation of privacy. We have taken care to avoid any biases or discriminatory outcomes in our research process. No personally identifiable information was used, and no experiments were conducted that could raise privacy or security concerns. We are committed to maintaining transparency and integrity throughout the research process.

## REPRODUCIBILITY STATEMENT

To ensure the reproducibility of our work, we have provided our source code in the supplementary materials. Upon the paper's acceptance, we will make the code publicly available on a GitHub repository. All experimental setups, including dataset details, incremental learning configurations, and hyperparameters, are described in section 4.1 and section B.5. The theoretical framework and its corresponding proofs that form the basis of our method are detailed in section 3.2 and section B.1. We believe these resources provide the necessary details for the research community to reproduce our findings.

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

## A   THE USE OF LARGE LANGUAGE MODELS

During the preparation of this manuscript, we utilized a Large Language Model (LLM) as a writing assistant. The role of the LLM was strictly limited to language polishing, which included improving grammar, clarity, and overall readability. The LLM did not contribute to the research ideation, experimental design, methodology, or analysis of the results. All authors have reviewed the final text and take full responsibility for the content of this paper.

## B   THEORETICAL ANALYSIS OF DYNAMIC BACKGROUND SUPERVISION

### B.1   FORGETTING RATE

For the forgetting rate in Definition 3.2, the average forgetting rate $\mathcal{E}_\tau(\theta)$ is defined as the arithmetic average of the forgetting rates of historical tasks:

$$\mathcal{E}_\tau(\theta) = \mathcal{L}_\tau^{val}(\theta) - \mathcal{L}_\tau^{val}(\theta_\tau^*)$$

Applying Taylor expansion to the above equation gives us:

$$\mathcal{E}_\tau(\theta) = (\theta - \theta_\tau^*)^\mathsf{T} \nabla \mathcal{L}_\tau^{val}(\theta_\tau^*) + \frac{1}{2} (\theta - \theta_\tau^*)^\mathsf{T} \mathbf{H}_\tau(\theta_\tau^*)(\theta - \theta_\tau^*) + O(\|\theta - \theta_\tau^*\|^3) \qquad (6)$$

In Equation (6):

- First order $(\theta - \theta_\tau^*)^\mathsf{T} \nabla \mathcal{L}_\tau^{val}(\theta_\tau^*)$: said loss function near the optimal parameters of linear change;

- Second order terms $\frac{1}{2} (\theta - \theta_\tau^*)^\mathsf{T} \mathbf{H}_\tau(\theta_\tau^*)(\theta - \theta_\tau^*)$: by Hessian Matrix $\mathbf{H}_\tau(\theta_\tau^*)$ said the local curvature of loss function;

- high order events $O(\|\theta - \theta_\tau^*\|^3)$: said the higher order nonlinear effects.

In Assumption 3.1: $\nabla \mathcal{L}_\tau(\theta_\tau^*) \leq \epsilon$, including $\epsilon$ is a limitless tends to zero (in the optimal parameter $\theta_\tau^*$, The gradient of the loss function has gone to zero), so the first-order term in the Taylor expansion is ignored, and the forgetting rate is dominated by the second-order term:

$$\mathcal{E}_\tau(\theta) = \frac{1}{2} (\theta - \theta_\tau^*)^\mathsf{T} \mathbf{H}_\tau(\theta_\tau^*)(\theta - \theta_\tau^*) + O(\|\theta - \theta_\tau^*\|^3) \qquad (7)$$

For two specific tasks $i, j$ and the optimal parameters $\theta_j^*$ on the tasks $j$ (assuming that the parameters vary within a range of $\delta$, That's $\|\theta - \theta_t^*\| \leq \delta$) (we've added the range of the parameter to our assumption). We find that based on Equation (7), this forgetting rate can be simplified to Equation (8), and the simplified formula is as follows:

$$\mathcal{E}_i(\theta_j^*) = \frac{1}{2}(\theta_j^* - \theta_i^*)^\intercal \mathbf{H}_i(\theta_i^*)(\theta_j^* - \theta_i^*) + \mathcal{O}(\delta^3)$$

$$= \frac{1}{2}\left(\sum_{\tau=i+1}^{j} \Delta_\tau\right)^\intercal \mathbf{H}_i(\theta_i^*)\left(\sum_{\tau=i+1}^{j} \Delta_\tau\right) + \mathcal{O}(\delta^3)$$

$$= \frac{1}{2}\left(\Delta_j + \sum_{\tau=i+1}^{j-1} \Delta_\tau\right)^\intercal \mathbf{H}_i(\theta_i^*)\left(\Delta_j + \sum_{\tau=i+1}^{j-1} \Delta_\tau\right) + \mathcal{O}(\delta^3)$$

$$= \underbrace{\frac{1}{2}\left(\sum_{\tau=i+1}^{j-1} \Delta_\tau\right)^\intercal \mathbf{H}_i(\theta_i^*)\left(\sum_{\tau=i+1}^{j-1} \Delta_\tau\right)}_{\mathcal{E}_i(\theta_{j-1}^*)} + \frac{1}{2}\Delta_j^\intercal \mathbf{H}_j(\theta_j^*)\Delta_j$$

$$+ \frac{1}{2}\left(\sum_{\tau=i+1}^{j-1} \Delta_\tau\right)^\intercal \mathbf{H}_i(\theta_i^*)\Delta_j + \frac{1}{2}\Delta_j^\intercal \mathbf{H}_i(\theta_i^*)\left(\sum_{\tau=i+1}^{j-1} \Delta_\tau\right) + \mathcal{O}(\delta^3)$$

$$= \mathcal{E}_i(\theta_{j-1}^*) + \frac{1}{2}\Delta_j^\intercal \mathbf{H}_i(\theta_i^*)\Delta_j + \left(\sum_{\tau=i+1}^{j-1} \Delta_\tau\right)^\intercal \mathbf{H}_i(\theta_i^*)\Delta_j + \mathcal{O}(\delta^3)$$

The Forgetting rate for task $i$ and optimal parameter $\theta_j^*$ can be expressed as Equation (8)

$$\mathcal{E}_i(\theta_j^*) = \mathcal{E}_i(\theta_{j-1}^*) + \frac{1}{2}\Delta_j^\intercal \mathbf{H}_i(\theta_i^*)\Delta_j + \left(\sum_{\tau=i+1}^{j-1} \Delta_\tau\right)^\intercal \mathbf{H}_i(\theta_i^*)\Delta_j + \mathcal{O}(\delta^3) \qquad (8)$$

In Equation (8), it consists of several parts: past forgotten rate: $\mathcal{E}_i(\theta_{j-1}^*)$, namely after the completion of the task $j-1$ for task $i$ forgotten; independent effects of the current parameter update: $\frac{1}{2}\Delta_j^\intercal \mathbf{H}_i(\theta_i^*)\Delta_j$, directly caused by the parameter update $\Delta_j$ of task $j$; interaction between historical and current updates : $\left(\sum_{\tau=i+1}^{j-1} \Delta_\tau\right)^\intercal \mathbf{H}_i(\theta_i^*)\Delta_j$, which reflects the nonlinear superposition effect of the parameter update sequence, and represents the inner product of the historical update and the current update. If the two directions are negatively correlated under the measure of $\mathbf{H}_i(\theta_i^*)$(e.g., orthogonal or reverse), forgetting may be alleviated. On the contrary, if the direction is consistent, the forgetting is aggravated.

Similarly, for the average forgetting rate $\bar{\mathcal{E}}_t(\theta_t^*) = \frac{1}{t-1}\sum_{\tau=1}^{t-1}\mathcal{E}_\tau(\theta_t^*)$, we can also simplify it through Equation (7)

$$\bar{\mathcal{E}}_t(\theta_t^*) = \frac{1}{t-1}\sum_{\tau=1}^{t-1}\left(\mathcal{E}_\tau(\theta_{t-1}^*) + \frac{1}{2}\Delta_t^\intercal \mathbf{H}_\tau(\theta_\tau^*)\Delta_t + \left(\sum_{o=\tau+1}^{t-1}\Delta_o\right)^\intercal \mathbf{H}_\tau(\theta_\tau^*)\Delta_t\right) + \mathcal{O}(\delta^3)$$

$$= \frac{1}{t-1}\left(\sum_{\tau=1}^{t-2}\left(\mathcal{E}_\tau(\theta_{t-1}^*)\right) + \frac{1}{2}\sum_{\tau=1}^{t-1}\Delta_t^\intercal \mathbf{H}_\tau(\theta_\tau^*)\Delta_t + \sum_{\tau=1}^{t-1}\left(\sum_{o=\tau+1}^{t-1}\Delta_o\right)^\intercal \mathbf{H}_\tau(\theta_\tau^*)\Delta_t\right) + \mathcal{O}(\delta^3)$$

$$= \frac{t-2}{t-1}\bar{\mathcal{E}}_{t-1}(\theta_{t-1}^*) + \frac{1}{2(t-1)}\Delta_t^\intercal\left(\sum_{\tau=1}^{t-1}\mathbf{H}_\tau(\theta_\tau^*)\right)\Delta_t + \frac{1}{t-1}\left(\underbrace{\sum_{\tau=1}^{t-1}(\theta_{t-1}^* - \theta_\tau^*)^\intercal \mathbf{H}_\tau(\theta_\tau^*)}_{v_t^\intercal}\right)\Delta_t$$

$$+ \mathcal{O}(\delta^3)$$

$$= \frac{1}{t-1}\left((t-2)\bar{\mathcal{E}}_{t-1}(\theta_{t-1}^*) + \frac{1}{2}\Delta_t^\intercal(\sum_{\tau=1}^{t-1}\mathbf{H}_\tau(\theta_\tau^*))\Delta_t + v_t^\intercal\Delta_t\right) + \mathcal{O}(\delta^3)$$

In the above simplification, each average forgetting rate has a specific $v$, which we denote as $v_t$, representing that it belongs to $\bar{\mathcal{E}}_t(\theta_t^*)$

$$\bar{\mathcal{E}}_t(\theta_t^*) = \frac{1}{t-1}\left((t-2)\cdot\bar{\mathcal{E}}_{t-1}(\theta_{t-1}^*) + \frac{1}{2}\Delta_t^\intercal(\sum_{o=1}^{t-1}\mathbf{H}_o^\star)\Delta_t + v_t^\intercal\Delta_t\right) + \mathcal{O}(\delta^3) \qquad (9)$$

From this equation, we can observe that the average forgetting rate for task $t$, $\bar{\mathcal{E}}_t(\theta_t^*)$, is influenced by three principal terms (ignoring the higher-order term $\mathcal{O}(\delta^3)$) that are averaged:

- $(t-2)\cdot\bar{\mathcal{E}}_{t-1}(\theta_{t-1}^*)$ relates to the average forgetting rate of the immediately preceding task.

- $\frac{1}{2}\Delta_t^\intercal(\sum_{o=1}^{t-1}\mathbf{H}_o^\star)\Delta_t$ captures the impact of the parameter change $\Delta_t$ in conjunction with the cumulative Hessian matrices from all prior tasks (from $o=1$ to $t-1$).

Building on these observations, a further hypothesis can be formulated: It is proposed that if the average forgetting rate for each task from $k=2$ up to $k=t-1$ is zero (i.e., the first two items in $\bar{\mathcal{E}}_k(\theta_k^*)$ is zero for all $k \in \{2,3,\ldots,t-1\}$), then for the current $t$-th task, the component $v^\mathrm{T}\Delta_t$ is also hypothesized to be zero.

We will use mathematical induction to prove this hypothesis. At first, we suppose the first two terms of $\bar{\mathcal{E}}_2(\theta_2^*)$ are both 0, then we can get $v_3 = 0$:

$$\bar{\mathcal{E}}_3(\theta_3^*) = \frac{1}{2}\left(1\cdot\bar{\mathcal{E}}_2(\theta_2^*) + \frac{1}{2}\Delta_3^\intercal(H_2(\theta_2^*) + H_1(\theta_1^*))\Delta_3 + \sum_{t=1}^2(\boldsymbol{\theta}_2 - \boldsymbol{\theta}_t)^\intercal\boldsymbol{H}_t^\star\boldsymbol{\Delta}_3\right).$$

$$= 0 + \frac{1}{2}\Delta_3^\intercal(\frac{1}{2}H_2(\theta_2^*) + \frac{1}{2}H_1(\theta_1^*))\Delta_3 + \frac{1}{2}\underbrace{\Delta_2^\intercal H_1^1}_{=0}\Delta_3 + \frac{1}{2}\underbrace{(\boldsymbol{\theta}_2^\intercal - \boldsymbol{\theta}_2)}_{=0}H_t^\star\Delta_3$$

Then, we suppose the first tow terms of $\bar{\mathcal{E}}_{t-1}(\theta_{t-1}^*)$ is zero, we find that:

$$v_t - v_{t-1} = \sum_{o=1}^{t-1}(\theta_{t-1}^* - \theta_o^*)^\intercal\mathbf{H}_o(\theta_o^*) - \sum_{o=1}^{t-2}(\theta_{t-2}^* - \theta_o^*)^\intercal\mathbf{H}_o(\theta_o^*)$$

$$= \sum_{o=1}^{t-2}(\theta_{t-1}^* - \theta_o^*)^\intercal\mathbf{H}_o(\theta_o^*) - \sum_{o=1}^{t-2}(\theta_{t-2}^* - \theta_o^*)^\intercal\mathbf{H}_o(\theta_o^*)$$

$$= \sum_{o=1}^{t-2}\left(\theta_{t-1}^* - \theta_o^* - \theta_{t-2}^* + \theta_o^*\right)^\intercal\mathbf{H}_o(\theta_o^*)$$

Since in $\bar{\mathcal{E}}_{t-1}(\theta_{t-1}^*)$, $\frac{1}{2}\Delta_{t-1}^\intercal\sum_{o=1}^{t-2}H_o(\boldsymbol{\theta}_o^*)\Delta_{t-1} = 0$, So we find that: $v_t - v_{t-1} = 0$.

Also we can get the conclusion that:

$$v_t - v_{t-1} = \Delta_{t-1}^\intercal\sum_{\tau=1}^{t-2}\mathbf{H}_\tau(\theta_\tau^*) \qquad (10)$$

We can say that if $\bar{\mathcal{E}}_\tau(\theta_\tau^*) = 0, \forall\tau < t$, then $\bar{\mathcal{E}}_t(\theta_t^*) = \frac{1}{2(t-1)}\Delta_t^\intercal\left(\sum_{i=1}^{t-1}\mathbf{H}_i(\theta_i^*)\right)\Delta_t$, which is show in Theorem 3.3 (1).

In Theorem 3.3 (2), we need to proof the statement: $\mathcal{E}_\tau(\theta_t^*) = 0, \forall\tau < t \iff \Delta_t^\intercal\left(\sum_{\tau'=1}^{t-1}\mathbf{H}_{\tau'}(\theta_{\tau'}^*)\right)\Delta_t = 0$. It is clear that when $\mathcal{E}_\tau(\theta_t^*) = 0, \forall\tau < t$, then $\bar{\mathcal{E}}_\tau(\theta_\tau^*) = 0, \forall\tau < t$, using conclusion in Theorem 3.3(1), we can get the result: $\Delta_t^\intercal\left(\sum_{\tau'=1}^{t-1}\mathbf{H}_{\tau'}(\theta_{\tau'}^*)\right)\Delta_t = 0$.

When $\Delta_t^\intercal\left(\sum_{\tau'=1}^{t-1}\mathbf{H}_{\tau'}(\theta_{\tau'}^*)\right)\Delta_t = 0$, using Assumption 3.1, all Hessian matrix $\mathbf{H}_\tau(\theta_\tau^*)$ is positive semi-definite, then we have: $\Delta_\tau^\intercal(\sum_{o=1}^{\tau-1}\mathbf{H}_o(\theta_o^*)) = 0, \forall\tau < t$, using Equation (10), we have: $v_2 \le v_3 \cdots \le v_t \le v_{t+1}$. In definition, $\bar{\mathcal{E}}_2(\theta_2^*)$ is zero, and $\bar{\mathcal{E}}_3(\theta_3^*) = \frac{1}{8}\Delta_3^\intercal(\sum_{o=1}^2\mathbf{H}_o\theta_o^*)\Delta_3 = 0$, we have $\bar{\mathcal{E}}_t(\theta_t^*) = 0$ which can say that $\mathcal{E}_\tau(\theta_t^*) = 0, \forall\tau < t$

### B.2 OTHER ZERO FORGETTING STRATEGY

#### B.2.1 ORTHOGONAL GRADIENT METHOD

We follow the setting of the paper Farajtabar et al. (2020), where $\mathcal{L}$ is a non-negative loss function(CE or BCE loss). We also assume that the symbol $f_\theta$ represents the parameter $\theta$ used by the model ($f_\theta^c(x)$ means one of the output's channel which is about class $c$) and $N_t$ represents the total amount of data used in the tasks from 1 to $t$.

In Orthogonal Gradient Method Farajtabar et al. (2020), they want to address catastrophic forgetting in continual learning by keeping the updates for a new task orthogonal to the gradient directions associated with previous tasks' predictions. Formally, equal to Equation (11).

$$\langle \Delta_t^i, \nabla_{\theta_{t-1}^*} f_{\theta_{t-1}^*}^c(x_\tau) \rangle = 0 \quad \forall c \in \mathcal{C}_{[1:t-1]}, x_\tau \in D_\tau, \tau < t, \tag{11}$$

where $\Delta_t^i$ denotes the the $i$-th step for update.

Due to the previous studies Schraudolph (2002), the hessian matrix of the loss can be decomposed as two other matrices: the outer-product Hessian and the functional Hessian, and at the optimum parameter for loss function, the functional Hessian is negligible Singh et al. (2021). So we can write the approximation of the Hessian matrix under the optimal parameters in task $\tau$:

$$\mathbf{H}_\tau(\theta_\tau^*) = \frac{1}{N_\tau} \sum_{i=1}^{N_\tau} \nabla_{\theta_\tau^*} f_{\theta_\tau^*}(x_i)(\nabla_f^2 \mathcal{L}_\tau(x_i, y_i))\nabla_{\theta_\tau^*} f(x_i)^\intercal$$

When the parameter update follows Equation (11), it essentially satisfies the condition that the binomial is 0 mentioned in our zero forgetting condition in Theorem 3.3.

### B.3 WHY PARAMETERS ISOLATION IS ZERO FORGETTING

We find that when the SPI strategy is used, the sum of the historical Hessian matrix($\sum_{\tau=1}^{t-1} \mathbf{H}_\tau(\theta_\tau^*)$) is similar to a semi-positive definite block diagonal matrix to its parameter subspace:

$$\sum_{\tau=1}^{t-1} \mathbf{H}_\tau(\theta_\tau^*) = \begin{pmatrix} \mathbf{H}_1(\theta_1^*) & \mathbf{0} & \cdots & \mathbf{0} & \cdots & \mathbf{0} \\ \mathbf{0} & \mathbf{H}_2(\theta_2^*) & \cdots & \mathbf{0} & \cdots & \mathbf{0} \\ \vdots & \vdots & \ddots & \vdots & \cdots & \mathbf{0} \\ \mathbf{0} & \mathbf{0} & \cdots & \mathbf{H}_{t-1}(\theta_{t-1}^*) & \cdots & \mathbf{0} \\ \vdots & \vdots & \vdots & \vdots & \vdots & \vdots \\ \mathbf{0} & \mathbf{0} & \mathbf{0} & \mathbf{0} & \mathbf{0} & \mathbf{0} \end{pmatrix}. \tag{12}$$

Where:

- According to Assumption 3.1, Each $\mathbf{H}_\tau(\theta_\tau^*)$ on the diagonal is a square matrix representing the Hessian for task $\tau$ and is a semi-positive matrix. The dimensions of $\mathbf{H}_\tau(\theta_\tau^*)$ correspond to the number of new parameters introduced for task $\tau$.

- The $\prime$ symbols represent zero, indicating that the Hessian components for parameters of different tasks are decoupled due to SPI strategy.

Just as expressed in Equation (12), the sum of the historical tasks' Hessian matrices $\sum_{\tau=1}^{t-1} \mathbf{H}_\tau(\theta_\tau^*)$ and the parameter update for new task $\Delta_t$ are not in the same subspace, and the product between them must be 0 which is the zero forgetting condition in Theorem 3.3.

### B.4 CASCADE MODELING

**Variable Definitions.** We first clarify the random variables used throughout this derivation:

- $Y_p \in \mathcal{C}_{[1:t]} \cup \{0\}$: the random variable representing the class label assigned to pixel $p$, where 0 denotes background.

- $Z_c \in \{0, 1\}$: a binary random variable indicating whether class $c$ is present in image $\mathbf{x}$. Formally, $Z_c = 1 \Leftrightarrow \exists\, p : Y_p = c$.

We start from the task-level factorization introduced in Equation (4):

$$P(Y_p = c \mid \mathbf{x}) = P(Y_p = c \mid \mathbf{x}, \mathcal{T}_t)\, P(\mathcal{T}_t \mid \mathbf{x}), \quad c \in \mathcal{C}_t,$$

where $\mathcal{T}_t$ denotes the task that first introduced class $c$. Under SPI, classes are disjoint across tasks: $\mathcal{C}_\tau \cap \mathcal{C}_t = \varnothing$ for all $\tau \neq t$.

**Lemma B.1** (Task Prior Decomposition). *For any image* $\mathbf{x}$,

$$P(\mathcal{T}_t \mid \mathbf{x}) = P\left(\bigvee_{c \in \mathcal{C}_t} Z_c = 1 \,\Big|\, \mathbf{x}\right) \approx \sum_{c \in \mathcal{C}_t} P(Z_c = 1 \mid \mathbf{x}), \tag{13}$$

*where* $P(Z_c = 1 \mid \mathbf{x})$ *is the probability that class* $c$ *exists somewhere in image* $\mathbf{x}$, *and the approximation holds when class co-occurrence within a single task is rare.*

*Proof.* Since classes in $\mathcal{C}_t$ are mutually exclusive and task $\mathcal{T}_t$ is active if and only if at least one class from $\mathcal{C}_t$ appears in $\mathbf{x}$, we have

$$P(\mathcal{T}_t \mid \mathbf{x}) = P\left(\bigvee_{c \in \mathcal{C}_t} Z_c = 1 \,\Big|\, \mathbf{x}\right) \approx \sum_{c \in \mathcal{C}_t} P(Z_c = 1 \mid \mathbf{x})$$

by the inclusion-exclusion principle (approximated when inter-class overlap is negligible). $\square$

**From Task-Level to Class-Level Factorization**   The task-level factorization necessarily reduces to a simpler class-level form that eliminates the problematic task prior $P(\mathcal{T}_t \mid \mathbf{x})$.

**Theorem B.2** (Cascade Factorization). *For any pixel* $p$, *class* $c$, *and image* $\mathbf{x}$:

$$P(Y_p = c \mid \mathbf{x}) = P(Y_p = c \mid \mathbf{x}, Z_c = 1) \cdot P(Z_c = 1 \mid \mathbf{x}). \tag{14}$$

*Proof.* We derive this using the law of total probability over the class existence variable $Z_c$:

$$P(Y_p = c \mid \mathbf{x}) = P(Y_p = c \mid \mathbf{x}, Z_c = 1) \cdot P(Z_c = 1 \mid \mathbf{x})$$
$$+ P(Y_p = c \mid \mathbf{x}, Z_c = 0) \cdot P(Z_c = 0 \mid \mathbf{x}).$$

The key insight: if class $c$ is not present in the image ($Z_c = 0$), then no pixel can be labeled as $c$. Therefore, $P(Y_p = c \mid \mathbf{x}, Z_c = 0) = 0$. This eliminates the second term, yielding:

$$P(Y_p = c \mid \mathbf{x}) = P(Y_p = c \mid \mathbf{x}, Z_c = 1) \cdot P(Z_c = 1 \mid \mathbf{x}).$$

$\square$

**Normalization over All Classes**   The final pixel-level prediction requires normalization over all candidate classes. Let $\mathcal{C}_{pred} = \{c : P(Z_c = 1|\mathbf{x}) \geq \alpha\}$ be the set of predicted classes after thresholding. The normalized probability is:

$$P(Y_p = c \mid \mathbf{x}) = \frac{P(Y_p = c \mid \mathbf{x}, Z_c = 1) \cdot P(Z_c = 1 \mid \mathbf{x})}{\sum_{c' \in \mathcal{C}_{pred} \cup \{0\}} P(Y_p = c' \mid \mathbf{x}, Z_{c'} = 1) \cdot P(Z_{c'} = 1 \mid \mathbf{x})}, \tag{15}$$

where $c' = 0$ corresponds to the background class with $P(Z_0 = 1|\mathbf{x}) = 1$ (background is always present).

**Implications for CogCaS Architecture**   Theorem B.2 proves that under SPI, the cascade factorization in Equation (14) is the unique probabilistically consistent decomposition. This directly motivates our two-phase CogCaS design:

- **Phase I**: Multi-label classifier estimates class existence probabilities $P(Z_c = 1 \mid \mathbf{x})$
- **Phase II**: Binary segmentation heads model conditional segmentation $P(Y_p = c \mid \mathbf{x}, Z_c = 1)$

This decomposition eliminates the ill-defined task prior, enables complete parameter isolation for zero-forgetting, and aligns with optimal Bayesian factorization principles.

## B.5 TRAINING AND INFERENCE

**CISS Training.** For every new task $T_t$, we append one multi-label classifier head and one class-specific binary segmentation head (Deeplab-v3's ASPP module + $1 \times 1$ conv) for each unseen class $c \in \mathcal{C}_t$, while all previous weights are frozen (*Strict Parameter Isolation*).

Furthermore, during the training stage, we manually construct near-out-of-distribution (near-OOD) data based on the data available for the current task to enhance the model's robustness. As illustrated in the Figure 5, this process is divided into two distinct stages: Phase I and Phase II. It is important to note that our model is trained separately in these two phases.

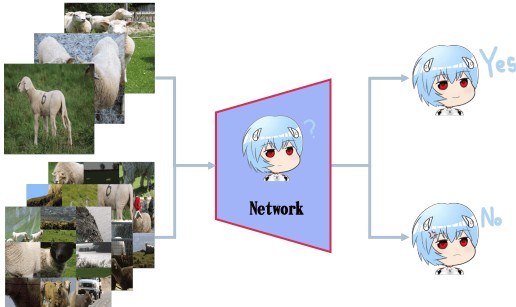

Figure 5: The figure illustrate near-ood samples and process.

**Joint Training.** In the joint training setting, we first train the classifier in Phase I. After the training for Phase I is complete, because we have access to the full dataset, we then select corresponding out-of-distribution data for each class-specific segmentation head in Phase II. This data is chosen from the dataset at a 1:1 ratio, and we ensure that for a specific class, its corresponding OOD samples do not contain any information about that class in any single pixel.

**Parameter Overhead.** Each additional class contributes $\approx 1.37$ M parameters (5.24 MB at FP32), corresponding to 3.1 % of a ResNet-101 backbone (44.5 M params) and 4.9 % of a Swin-T backbone (28 M params); even for the larger Swin-B (88 M params) the overhead per class is merely 1.6 %.

**Loss Functions.** Classification and segmentation are optimised separately and then summed:

$$\mathcal{L}_{\text{cls}} = \tfrac{1}{|\mathcal{B}|} \sum_{x \in \mathcal{B}} \sum_{c \in \mathcal{C}_{1:t}} \text{BCE}\big(P(c \mid x),\, y_c\big)$$

$$\mathcal{L}_{\text{seg}} = \tfrac{1}{|\mathcal{B}|} \sum_{x \in \mathcal{B}} \sum_{c \in \mathcal{C}_t} \Big[ \alpha \, \text{Focal}(M_c, \hat{M}_c) + \beta \, \text{Dice}(M_c, \hat{M}_c) \Big]$$

$$\text{Focal}(M_c, \hat{M}_c) = -\tfrac{1}{N} \sum_{i=1}^{N} \Big[ \alpha \, M_{c,i}(1 - \hat{M}_{c,i})^\gamma \log(\hat{M}_{c,i}) + (1 - \alpha)(1 - M_{c,i})(\hat{M}_{c,i})^\gamma \log(1 - \hat{M}_{c,i}) \Big]$$

$$\text{Dice}(M, \hat{M}) = \frac{2 \sum_i M_i \hat{M}_i + \varepsilon}{\sum_i M_i + \sum_i \hat{M}_i + \varepsilon}, \quad \varepsilon = 10^{-6}$$

**Notation.** $\mathcal{B}$: mini-batch; $\mathcal{C}_{1:t}$: all classes learned up to task $t$; $\mathcal{C}_t$: classes introduced at task $t$; $P(c \mid x)$: predicted presence probability for class $c$; $y_c \in \{0, 1\}$: image-level label; $M_c, \hat{M}_c$: predicted / ground-truth masks; $i$: spatial index.

The full objective is $\mathcal{L} = \mathcal{L}_{\text{cls}} + \lambda \, \mathcal{L}_{\text{seg}}$ with $\lambda = 1$.

**Optimisation Schedule.** Epoch 1 trains only the new classifier heads; the remaining epochs finetune both classifier and segmentation heads. SGD (momentum 0.9, weight-decay $10^{-4}$), batch size 20, initial LR $5 \times 10^{-3}$ with cosine decay is used.

---

**Algorithm 1** Inference Pipeline (per image)

---

**Require:** backbone $\Phi$, multi-label head $G$, binary heads $\{H_c\}$
1: $F \leftarrow \Phi(x)$      ▷ shared feature map
2: $\mathbf{p} \leftarrow G(F)$      ▷ class-presence probabilities
3: $\mathcal{C}_{\text{pred}} \leftarrow \{\, c \mid \mathbf{p}[c] > 0.5 \,\}$      ▷ or top-$k$
4: **for all** $c \in \mathcal{C}_{\text{pred}}$ **do**
5:      $M_c \leftarrow H_c(F)$      ▷ binary mask for class $c$
6: **end for**
7: $M_{\text{final}} \leftarrow \mathcal{U}\big(\{M_c\}_{c \in \mathcal{C}_{\text{pred}}}\big)$      ▷ mask fusion
8: **return** $M_{\text{final}}$

---

**Inference.** Because only $|\mathcal{C}_{\text{pred}}|$ segmentation heads are activated, inference cost scales with the number of present classes rather than the total number of learned classes.

## B.6 COMPARISON WITH LARGE-MODEL-BASED METHODS

We compare CogCaS with recent large-model-based CISS methods in table 7. SAM-based methods (e.g., DecoupleCSS) leverage a foundation model with $\sim$632M parameters pre-trained on 11M images, representing a fundamentally different resource regime compared to our standard backbones (ResNet-101: 44.5M; Swin-L: 197M). Our "Cognitive Cascade" architecture and theoretical zero-forgetting guarantee are orthogonal to backbone choice and can be readily integrated with SAM or similar models. CogCaS offers a favorable balance between performance and efficiency, particularly for resource-constrained deployment scenarios.

Table 7: Comparison with large-model-based CISS methods. $\star$: SAM-based ($\sim$632M params). $\ddagger$: Mask2Former-based.

| Method | Setting | Old | New | All |
|---|---|---|---|---|
| *PASCAL VOC 2012* | | | | |
| DecoupleCSS$^\star$ | 19-1 | 82.9 | 83.7 | 84.0 |
| DecoupleCSS$^\star$ | 15-1 | 83.8 | 82.1 | 83.4 |
| **Ours$^\circ$** | 15-1 | 75.5 | 71.4 | 74.4 |
| *ADE20K* | | | | |
| DecoupleCSS$^\star$ | 100-10 | 58.2 | 52.0 | 56.9 |
| DecoupleCSS$^\star$ | 100-5 | 57.5 | 55.6 | 56.9 |
| CIT-M2Former$^\ddagger$ | 100-10 | 56.9 | 38.2 | 50.7 |
| CIT-M2Former$^\ddagger$ | 100-5 | 55.5 | 32.2 | 47.7 |
| SimCIS$^\ddagger$ | 100-10 | 49.7 | 27.4 | 42.3 |
| SimCIS$^\ddagger$ | 100-5 | 46.7 | 22.8 | 38.7 |
| **Ours$^\circ$** | 100-10 | 49.2 | 44.0 | 47.4 |
| **Ours$^\circ$** | 100-5 | 49.5 | 43.6 | 47.3 |

