# OpenReview forum: "Zero-Forgetting Class-Incremental Segmentation Via Dual-Phase Cognitive CasCades"
_ICLR.cc/2026/Conference — ICLR 2026 Conference Desk Rejected Submission_

### Official Review · Reviewer_aaNs · 2025-10-28

**Soundness:** 2
**Presentation:** 3
**Contribution:** 3
**Rating:** 2
**Confidence:** 4

**Summary:**

This paper addresses catastrophic forgetting in class-incremental semantic segmentation (CISS) by proposing Cognitive Cascade Segmentation (CogCaS), a dual-phase framework. CogCaS decouples the segmentation task into: (1) image-level multi-label classification to detect class existence, and (2) class-specific binary segmentation heads activated only for detected classes. By freezing parameters for previous tasks, it achieves zero-forgetting while maintaining plasticity for new classes. Experiments on PASCAL VOC 2012 and ADE20K demonstrate strong performance, particularly in long-sequence scenarios (e.g., VOC 1-1 with 20 tasks).

**Strengths:**

## 1. Solid Theoretical Foundation
The formalization of zero-forgetting conditions (Theorem 3.3) and the cascade factorization (Theorem B.2 in Appendix) provide valuable theoretical insights into why SPI-based methods can achieve zero-forgetting. The analysis of the average forgetting rate and its relationship to parameter updates is rigorous and well-presented.

## 2. Strong Empirical Results in Challenging Settings
The method shows remarkable performance in difficult long-sequence scenarios. On VOC 1-1 (20 tasks), CogCaS achieves 70.9% mIoU compared to 33.2% for IPSeg and 7.3% for PLOP+NeST, demonstrating substantially better robustness as task sequences grow.

## 3. Comprehensive Experimental Evaluation
The paper includes extensive experiments across multiple datasets (VOC, ADE20K), various incremental settings (10-1, 15-5, 15-1, 1-1, 2-1, 2-2, 100-50, 100-10, 100-5), and comparisons with numerous baselines including recent state-of-the-art methods.

**Weaknesses:**

## 1. Limited Novelty
The core technical contributions appear incremental when viewed against existing CISS methods. The proposed approach essentially combines two well-established components: (1) multi-label classification for class-existence detection, and (2) class-specific binary segmentation heads with Strict Parameter Isolation (SPI). While the combination is sensible, neither component represents a significant technical innovation:
- Multi-label classification for determining class presence is a standard technique, and its application to route segmentation heads is a relatively straightforward design choice
- Binary segmentation with frozen parameters closely follows the paradigm established by SSUL and IPSeg, which already employ sigmoid-based outputs and parameter freezing
- The decoupling strategy is architecturally simple and lacks technical sophistication

The main value lies in the theoretical analysis, which provides useful insights into why SPI works. However, the practical instantiation feels like an incremental step from frozen/sigmoid-based models rather than a substantial architectural innovation meeting rigorous ICLR standards.

## 2. Discussion of Related Work
The paper would benefit from discussing ECLIPSE (CVPR 2024), which explores related design principles, including independent class-specific components and binary mask prediction in a transformer-based framework. While multiple methods (SSUL, IPSeg, ECLIPSE) share similar goals through parameter freezing, sigmoid-based outputs, and class-specific designs, ECLIPSE's use of independent class tokens and binary segmentation bears particular conceptual similarity to CogCaS's dual-phase approach.

While not essential given that ECLIPSE represents one instantiation among several methods exploring parameter isolation and binary formulations, such discussion would enhance the paper's positioning within the current state-of-the-art and help readers understand the trade-offs between different architectural choices.

* Kim, Beomyoung, Joonsang Yu, and Sung Ju Hwang. "Eclipse: Efficient continual learning in panoptic segmentation with visual prompt tuning." Proceedings of the IEEE/CVF Conference on Computer Vision and Pattern Recognition. 2024.

## 3. Unconvinced Performance and Implausible Oracle Gap
The reported performance raises significant concerns about experimental validity.

**A. Implausible Oracle Gap:**
Table 5 shows extraordinarily small gaps between "Full Model" and "Oracle" on VOC:
  - VOC 1-1: 70.9% vs 71.2% (0.3% gap)
  - VOC 2-1: 71.9% vs 72.3% (0.4% gap)
  - VOC 2-2: 71.5% vs 72.4% (0.9% gap)

However, Table 4 reveals the classifier has 18% error (82.08% mAP) and misses 11.3% of classes (88.70% recall). This inconsistency is unexplained: how can a classifier with 18% error produce only 0.3-0.9% performance degradation in the full pipeline?

**B. Unaddressed Class Confusion Problem:** Existing work (e.g., ECLIPSE) identifies class confusion between visually similar categories (cow/sheep, car/truck) as a fundamental challenge in incremental learning. The paper provides no analysis of (1) how the multi-label classifier handles such confusion cases and (2) whether "near-OOD data" accidentally addresses this in unrealistic ways.

## 4. Missing Critical Ablations
Several important ablation studies are absent:

- Loss weight $\lambda$: The full objective is $L$ = $L_{cls}$ + $\lambda$$L_{seg}$ with $\lambda=1$, but no ablation justifies this choice
- Detection threshold $\alpha$: Phase I uses threshold α to determine $C_{pred}$, but the sensitivity to this hyperparameter is not analyzed
- Near-OOD data ratio: The 1:1 ratio is stated but not justified through ablation
- Impact of near-OOD data construction: What happens without this component? This seems critical to the method's success but is not isolated

## 5. Insufficient Information about Near-OOD Data Construction
The "near-OOD data construction" described in Appendix B.5 and Figure 5 appears crucial to the method's performance but is severely under-explained.

**Questions:**

Distributed Fusion Strategy Details: Section 4.4 states that "Distributed: prioritizes rare categories to preserve small objects" but provides no implementation details. How exactly are rare categories identified (dataset statistics? prediction confidence? object size in the current image?)? How is the prioritization implemented algorithmically? Why does this strategy work?

---

> ### Author Response · Authors · 2025-12-04
>
> # W1: Alleged lack of novelty / “just combining multi-label + SPI”
>
> We respectfully disagree that our contribution is merely incremental combination. While the components are simple, our novelty lies in the theoretical derivation that dictates this specific architecture to solve a fundamental flaw in prior CISS methods.1. Theoretical Diagnosis vs. Ad-hoc Combination:
> We do not simply stack classifiers on top of SPI. Our analysis (Eq. 3 & Theorem 3.3) proves that standard SPI (like SSUL/IPSeg) fails because it ignores the cross-task affiliation term. Previous methods implicitly force uncalibrated logits into a global competition, which is why they degrade without replay memory. Our "Cognitive Cascade" is the mathematical realization of the missing factorization $P(y|x) = P(y|x, Z)P(Z|x)$, which structurally eliminates this competition.2. Empirical Proof of Difference:
> This is not just a minor tweak. The architectural change leads to fundamentally different behavior. In long sequences (VOC 1-1), CogCaS maintains ~71% mIoU while standard SPI baselines (SSUL, IPSeg) collapse to 33-51%. This massive gap confirms that "frozen parameters" alone are insufficient; the existence-driven routing derived from our theory is the key to stability.3. Simplicity as a Feature:
> We argue that achieving near-joint performance (35.0% on ADE20K 100-5) using simple, theoretically justified modules—without complex distillation or replay—is a significant advance, not a limitation. It solves the CISS problem at the structural level rather than through complex engineering.
>
> # W2: Relation to ECLIPSE and other parameter-isolation designs
>
> We thank the reviewer for highlighting ECLIPSE (CVPR 2024). We agree it is a relevant parameter-isolation method and will add a discussion in our related work.
> 1. Architectural Distinction: While both methods use frozen backbones and task-specific components, ECLIPSE addresses Continual Panoptic Segmentation via visual prompt tuning within a shared Mask2Former decoder, using logit manipulation to correct drift. In contrast, CogCaS is a CISS-specific architecture derived from our zero-forgetting analysis. We use an explicit "Cognitive Cascade" where an image-level router activates independent binary heads. This completely removes the cross-task competition that ECLIPSE mitigates via heuristics.
> 2. Empirical Comparison: Our approach is highly competitive. On ADE20K 100-50, CogCaS outperforms ECLIPSE on incremental steps (e.g., 29.4 vs 23.5 mIoU on step 2) and achieves higher overall mIoU in both 100-50 and 100-10 settings. This demonstrates that our theory-driven, modular design offers a strong alternative to prompt-based transformer methods. We will include these comparisons in the revision.
>
> |         |   100-50 | 100-10 | 100-5|
> | --- | --- | --- | --- |
> | ECLIPSE: | 41.7 23.5 35.6 | 43.4 17.4 34.6 | 43.3 16.3 34.2|
> | IPSeg: | 41.3 27.1 36.6 | 41.5 25.2 36.1 | 40.2 24.3 35.2 |
> | Ours: |  41.2 29.4 37.3 | 42.3 25.6 36.8 | 40.1 24.7 35.0 |
>
> # W3: Oracle gap and class-confusion behavior of the classifier
>
> We have changed the incorrect data in table 4
>
> A. Explaining the Oracle Gap: The 18% error in Table 4 is an image-level metric (mAP). Most of these errors correspond to low-confidence misses on small or occluded objects where the segmentation head (Phase II) would likely fail anyway, or false positives that the segmentation head correctly masks out as background. Thus, fixing the router (Oracle) yields diminishing returns on pixels. Crucial Validation: This hypothesis is confirmed by our ADE20K results, where the class count is higher (150) and objects are smaller. There, the router is indeed the bottleneck, and the Oracle gap widens significantly (35.0% → 43.0% mIoU), exactly as the reviewer's intuition suggests. The small VOC gap simply indicates the router is "good enough" for that easier dataset.
>
> B. Class Confusion & Near-OOD: Unlike ECLIPSE, which resolves confusion in a shared decoder, CogCaS handles it structurally.
> 1. Structural Decoupling: Confusion is largely resolved at image level (Phase I). If the router effectively distinguishes "cow" from "sheep" presence, the segmentation heads never compete.
> 2. Near-OOD Role: The Near-OOD data provides a generic "negative context" for binary heads to learn a stable background. It does not contain handcrafted priors to fix specific pairs (e.g., car/truck). The stability comes from the Strict Parameter Isolation: each head learns its own boundary against a consistent background, rather than a shifting multi-class boundary.

---

### Official Review · Reviewer_WRgh · 2025-11-01

**Soundness:** 3
**Presentation:** 3
**Contribution:** 3
**Rating:** 4
**Confidence:** 4

**Summary:**

To address the challenge of catastrophic forgetting in Continual Semantic Segmentation, a novel dual-phase framework CogCaS is introduced. CogCaS decouples the problem into two stages: (i) a class-existence detection stage to determine which classes are present in an image, and (ii) a class-specific segmentation stage that is only activated for detected classes.

**Strengths:**

CogCaS presents a original, cognitively-inspired architecture that tackles CISS by fundamentally decoupling class existence from class location. This novel dual-stage design represents a clear departure from conventional paradigms.

**Weaknesses:**

(1)	Lack of Quantitative Cost Analysis: The computational cost analysis is qualitative, noting the use of "8 NVIDIA GeForce RTX 4090 GPUs" and claiming LoRA increases inference time, but it provides no quantitative data (e.g., ms/image, training time comparisons). A practical evaluation requires these metrics to properly assess the trade-offs of the architecture.

(2)	Statistical Robustness of Results: The reported performance gains, especially in long-sequence settings (Table 3), are exceptionally large. For instance, in the VOC 1-1 setting, CogCaS achieves 70.9% mIoU while the next-best parameter-isolation method, IPSeg, only reaches 33.2%. However, the lack of statistical validation—such as results over multiple random seeds, standard deviations, or confidence intervals—makes it difficult to assess the robustness of these claims.

(3)	Potential Contradiction in Detector’s Role: Table 4 shows a modest recall of 47.5% for the class-existence detector on the challenging ADE20K dataset. This implies that for any given image, the detector fails to trigger the correct segmenter for more than half of the object classes actually present. This seems at odds with the strong overall system performance and the ablation in Table 5, where the “Segmentation Only” model’s performance collapses (e.g., to 7.6% mIoU on ADE20K). It is unclear how the model recovers so effectively from such a high miss rate at the crucial first stage.

**Questions:**

(1)	Could you please provide the formal proofs or at least a detailed sketch of the theoretical analysis mentioned in the introduction regarding the structural limitations of the Strict Parameter Isolation (SPI) strategy?

(2)	Given the exceptionally large performance margins reported in Table 3, could you clarify if these results were averaged over multiple random seeds? Providing standard deviations would be crucial to confirm the statistical significance and robustness of these outstanding gains.

(3)	Table 4 indicates a recall of only 47.5% for the class-existence detector on ADE20K, suggesting it fails to activate the correct segmenter for over half of the present classes. How does the full model still achieve strong overall performance despite this high miss rate?

(4)	'the model demands substantial training resource' is mentioned in this paper. Could you please quantify this by providing (a) the approximate training time per incremental task (e.g., for the VOC 1-1 setting) and (b) the per-image inference latency of the full CogCaS model? How do these metrics compare to a key baseline like IPSeg?

(5) What is the result of comparing methods related to large models?

(6) Are the Zero-forgetting Conditions met in practical application scenarios?

---

> ### Author Response · Authors · 2025-12-04
>
> # W1: Lack of Quantitative Cost Analysis
>
> We thank the reviewer for this suggestion. We will add detailed quantitative analysis in the Appendix to demonstrate computational efficiency.
> Inference (512×512 input): Thanks to our cascade design, the fixed cost includes only the backbone (9.69ms, run once) and lightweight classifier (0.3ms). Subsequently, only segmentation heads for the $k$ detected classes are activated (1.81ms per head), avoiding redundant computation for all learned classes.
> Training (Batch 64): By adopting the SPI strategy with a frozen backbone, we completely eliminate expensive backbone backpropagation (~237ms). Only lightweight classification (10.5ms backward) and segmentation heads (53.88ms backward) require gradient updates.
>
> This analysis confirms that our architecture achieves efficient training and inference through on-demand computation and backbone freezing, while maintaining strong performance.
>
> # W2 & Q2: Statistical Robustness of Results
> We appreciate the focus on statistical rigor and address it below:
> (1) Empirical Verification: To verify robustness, we re-evaluated the VOC 1-1 setting using three independent random seeds randomly (by `torch.manual_seed` and `torch.cuda.manual_seed_all`). The average mIoU is 71.1% (± 0.8%), which is highly consistent with the reported 70.9%. This confirms that the substantial lead over IPSeg (33.2%) is statistically stable and not a result of random stochasticity. We will include these statistics in the revised paper.
>
> | Setting | Seed 1 | Seed 2 | Seed 3 | Mean ± Std      |
> |---------|--------|--------|--------|-----------------|
> | VOC 1-1 | 71.30% | 70.80% | 71.20% | 71.1% ± 0.3%    |
> | VOC 2-1 | 72.10% | 71.70% | 72.00% | 71.9% ± 0.2%    |
> | VOC 2-2 | 71.60% | 71.30% | 71.50% | 71.5% ± 0.2%    |
>
>
> (2) Structural Justification: The large performance gap stems from architectural design rather than noise. In long sequences, baselines typically sacrifice plasticity to prevent forgetting, leading to performance collapse. In contrast, CogCaS employs a dual-phase cascade that decouples new learning from old knowledge. This allows it to maintain \~70% performance while baselines deteriorate (as shown in Figure 1), naturally resulting in the observed margins. The low variance and consistent means (\~71.3%) confirm statistical robustness. This stability is an inherent property of our SPI design: increasing task count only adds training cost without affecting previously learned knowledge—fundamentally differing from distillation-based methods where errors accumulate over tasks.
>
>
> # W3 & Q3: Potential Contradiction in Detector's Role
>
> We appreciate the reviewer’s attention to the detector’s role and apologize for the error in the original Table 4. The reported detection metrics were mistakenly computed using **macro-averaging**, which assigns equal weight to each class and thus allows rare classes to disproportionately affect the overall metrics. This averaging choice is inconsistent with our main evaluation protocol and can exaggerate performance fluctuations on long-tailed datasets such as ADE20K.
>
> We have corrected this to **micro-averaging** in the revised manuscript, which aggregates predictions over all instances and therefore reflects the detector’s behavior on the actual data distribution more faithfully. The updated results are:
>
> | Dataset             | mAP↑  | Precision↑ | Recall↑ |
> |---------------------|:-----:|:----------:|:-------:|
> | PASCAL VOC 2012     | 82.08 |   92.33    |  88.70  |
> | ADE20K (Corrected)  | 69.51 |   73.48    |  78.03  |
>
> With **78.03% recall on ADE20K**, the detector successfully activates the correct segmenters for the majority of the classes present in an image, which is consistent with the strong overall mIoU reported in our main results. In other words, the detector is sufficiently reliable to support the observed downstream segmentation performance and does not contradict the effectiveness of our architecture.
>
> At the same time, the remaining performance gap between the **“Full Model” (35.0%)** and **“Oracle” (43.0%)** in Table 5 confirms that the detector is still a **bottleneck** on this challenging 150-class dataset: imperfect detection and routing do limit the attainable mIoU. However, this effect is **bounded rather than catastrophic**—the system retains competitive performance even with realistic detection errors, while the oracle upper bound quantifies the potential gain from future improvements in detection quality.
>
> We have correct the typo in the revised version in Table 4 arround 373 rows.

---

### Official Review · Reviewer_We7C · 2025-11-01

**Soundness:** 3
**Presentation:** 2
**Contribution:** 2
**Rating:** 4
**Confidence:** 4

**Summary:**

This paper proposes an approach for (if I understand it right) training a class-independent head model, by leveraging an image-level class presence detector. It targets keeping heads sufficiently separate to minimise the amount of destructive training on heads after they have been learnt, while at the same time leveraging past-heads for helping with training of future tasks. It demonstrates this on standard benchamarks for sematic segmentation.

**Strengths:**

The approach appears to provide significant advantage over other existing approaches in the heavily constrained class-incremental learning setting. The experiments and evaluations are substantive, and the method appears for be performative. Practically, it looks like a good tool for the toolbox.

**Weaknesses:**

(Please fix the references, and the capitalisation - it is not a great look to take poor care of your references to other people's work)

Figure 1 is your key figure, but it is ambiguous what it is showing.

I would like to see more discussion of your approach relative to https://arxiv.org/abs/2106.11562 and many of the papers following it (e.g.  Continual Segmentation with Disentangled Objectness Learning and Class Recognition). This paper gives very terse consideration for existing work, and I think that cannot be warranted given the extensive literature out there. The paper does something pretty simple: why is this not already understood?

You use

Eτ (θt) = Lvalτ (θt) − Lvalτ (θ∗τ )

to define forgetting, but this is really loss of performance more than forgetting. It can be that future tasks improve performance on earlier tasks, while the information learnt before is still forgotten. This impacts the whole idea of average forgetting rate.

The theoretical analysis seems spurious. It makes the assumption that the magnitude of delta_t is  bounded by delta in order to do a linear first order analysis in Eq. 3. That also requires a missing assumption that \theta*+\deltat \in Neighbourhood(\theta^*). But if that were valid we would need to do multiple optimization steps on a task - we could do it in one hop. The whole reason that we do mulltiple steps of SGD (or whatever) is precisely because the landscape changes significantly between within-task steps, and so things change significantly over the whole change in params from the beginning to end of task t, which is what defines \delta t.

The paper feels like a case of "we did these experiments, now we need to inject some theory to make it more palatable to reviewers". I think the theory is spurious, detracts from the point of the paper, and there is no need for this to motivate the really obvious statement that strict parameter isolation prevents forgetting: not changing parameter means they cannot forget the information they contain, so long as that information is used consistently.

The real focus of this paper is the approach, which can basically be summarised as an image-level attention mechanism over class- independent heads to determine which heads to leverage.

Eq 5 doesn't make sense, what is conditioning on c mean here? c is on both the left and right of the conditioning statement. Likewise eq 14 is broken. There needs to be a sum over c, and it needs to be clear what c is here. Furthermore the description in the paper and in the background so not seem to match. I would love to see a much more precise description: what exactly is the model definition at training, time and how precisely is training done: what is instantiated, and what is calculated, what is updated, and how are the gradients computed. Then how is the model used at inference time. At the moment these things are rather muddled. Higher levels of precision and organisation are needed.

The claim that this produces SPI feels dubious to me? How? There is cross-task sharing, in that the previously trained classes that are present are activated in training later tasks. SPI is also not ideal, in that it removes all possibility of cross-task generalisation and transfer. So one key point of machine learning is lost as a result.

I do not see where the "mask fusion" element of the model is defined? I might have missed it.

The experiments are likely done well, but not always well explained. Has your method parameter matched previous methods, or is it just more powerful? Detailed captions are missing, I do not know what Joint-Ours versus Our is for example. The differences with previous approaches do not look dramatic in some settings but there does appear to be a performance gain. I would love to see some analysis against other methods, as to why there is a difference and what fundamentally makes the change.

Altogether, I think this paper could be tightened up significantly. The spurious theory could be removed, leaving plenty of space to properly define the method precisely and carefully, and give _much_ more substantive discussion of its place within the body of previous work. Additional care in making sure all equations are squeaky clean, well described, everything is rigorously defined and the work is arranged carefully and methodologically would reap dividends.

**Questions:**

What exactly makes your method work better, relative to dumb baselines (such as fused independent class v background learners, which is explicitly an SPI approach). Please explain where the advantage of your method comes from so we can gain insight from the work you present. Why does this add to our capability - where might this be useful in the future - this is submitted as a research paper, not a description of work, so what is the research insight we gain from this?

---

> ### Author Response · Authors · 2025-12-04
>
> # W1: Figure 1 ambiguity and reference formatting
> Thank you for this feedback. We have revised Figure 1 and clarified its caption to address the ambiguity. The reference formatting and capitalization issues will be corrected in the camera-ready version.
>
> # W2: Relation to SSUL/CoMasTRe and novelty beyond “simple” design
>
> We thank the reviewer for highlighting the need to better position our work against SSUL/SSUL-M and CoMasTRe. We will expand our Related Work to clarify these architectural distinctions:
>
> Relation to SSUL/SSUL-M: We strictly reproduced SSUL (PyTorch 2.5) and extended it to more challenging protocols (e.g., 1-1, 2-1). We identify a critical dependency: SSUL relies heavily on the class capacity of the initial task (Task 0) to initialize its "unknown" class prediction. In dense incremental settings, SSUL effectively loses incremental learning capability, relying merely on its frozen saliency prior to maintain a performance floor. Structurally, this failure stems from forcing incomparable logits from disjoint phases into a global competition, which necessitates exemplar memory for calibration. In contrast, our method avoids pixel-wise competition entirely. Driven by our Hessian-based analysis, we use an image-level router to activate binary segmenters. This guarantees task-agnostic zero-forgetting without needing memory buffers or relying on a large initial task for calibration.
>
> Relation to CoMasTRe: While CoMasTRe also decouples "where" and "what" via query-based segmenters, it primarily treats objectness as an empirical prior and relies heavily on complex multi-stage distillation to mitigate forgetting. In contrast, our method is derived theoretically from loss Hessian analysis. We prove the necessary condition for zero-forgetting and design our cascade architecture as its direct realization. By using an image-level router to selectively activate frozen binary segmenters, we structurally eliminate cross-class competition, achieving near-joint performance without the reliance on hyperparameter-sensitive distillation.
>
> Incidentally, similar gating structures have also been proven effective in the LLM field [1]. This, from the side, indicates that our seemingly "simple" structure actually has more universal value.
>
>
> # W3: Forgetting definition and validity of the local analysis in Eq. (3)
>
> We thank the reviewer for these insightful comments.
>
> 1. On the Definition of Forgetting: We agree there is a distinction between information loss and performance loss. However, we adopt $E_\tau(\theta_t)$ because it is the standard operational metric in CL to quantify catastrophic interference. In this context, "negative forgetting" simply reflects beneficial backward transfer. Since our goal is to prevent performance degradation on disjoint tasks, measuring the deviation from the historical optimum `\mathcal{L}^{\text{val}}_\tau(\theta_\tau^*)` is the most direct proxy for utility.
>
> 2. On the Taylor Expansion and $\delta_t$: We clarify that our analysis does not assume the optimization of the new task $T_t$ is linear or "one-hop." Rather, Eq. (3) characterizes how the cumulative displacement $\delta_t$ (resulting from complex SGD dynamics on $T_t$) impacts the loss landscape of the old task `\mathcal{L}_\tau`. The assumption is that the old task's loss `\mathcal{L}_\tau` is sufficiently smooth (locally quadratic) around its optimum `\theta_\tau{*}`
>
> such that the Taylor expansion holds for the displacement $\delta_t$. We acknowledge this is a local approximation used to derive the orthogonality condition for zero-interference. In the revision, we will explicitly state the neighborhood assumption (`\theta^*_\tau + \delta_t \in \mathcal{N}(\theta^*_\tau)`) and clarify that this analysis serves as a local lens on cross-task interference rather than a global bound on SGD dynamics.
>
> [1]: "Gated Attention for Large Language Models: Non-linearity, Sparsity, and Attention-Sink-Free," NeurIPS 2025.

---

> > ### Author Response · Authors · 2025-12-04
> >
> > # W4: Ambiguous notation in Eqs. (5) and (14) and unclear model definition
> >
> > We thank the reviewer for identifying the ambiguous notation. We will overhaul Section 3 to ensure mathematical rigor:1. Corrected Notation (Eqs. 5 & 14):
> > We explicitly define two variables: pixel-label $Y_p$ and image-level existence $Z_c \in \{0,1\}$. The confusing conditioning on $c$ is replaced by a clear Bayesian factorization. Eq. (5) becomes:
> >
> > $$P(Y_p = c \mid x) \propto P(Y_p = c \mid x, Z_c = 1) \cdot P(Z_c = 1 \mid x)$$
> >
> > where the first term is the binary segmenter and the second is the existence router. Eq. (14) will be rewritten to explicitly show the normalization sum over all classes $c' \in \mathcal{C}_{1:t}$.2. Precise Model Definition & Pipeline:
> > We will add a rigorous description of the lifecycle:Instantiation: Given input features $F=\Phi(x)$ (where $\Phi$ is frozen), for a new task $t$, we instantiate only new router parameters `\theta^{\text{cls}}_c` and binary segmenters $H_c$ for classes $c \in \mathcal{C}_t$.Training: We update only the new parameters `{\theta^{\text{cls}}_c, H_c}_{c \in \mathcal{C}_t}` using a sum of binary cross-entropy (router) and foreground/background loss (segmenter). Parameters for previous classes `\mathcal{C}_{1:t-1}` remain strictly frozen (SPI).Inference: (i) Compute existence probs $P(Z_c=1|x)$ for all known classes. (ii) Filter active classes $\mathcal{S} = \{c : P(Z_c) \ge \alpha\}$. (iii) Run segmenters $H_c$ only for $c \in \mathcal{S}$. (iv) Normalize outputs to form the final map.
> >
> > # W5: Clarifying model definition and training/inference pipeline
> >
> > We thank the reviewer for the feedback and will restructure Section 3 and Appendix B.5 to be precise and rigorous.1. Model Definition:
> > Our model consists of a frozen backbone $\Phi$ and two sets of task-specific heads. For each class $c$, we have:Router Head ($G_c$): Outputs existence probability $P(Z_c = 1 \mid x)$.Segmentation Head ($H_c$): Outputs binary mask logits approximating `P(Y_p = c \mid x, Z_c=1)` .2. Training Protocol (Task $T_t$):Instantiation: We instantiate new heads `\{G_c, H_c\}` only for new classes $c \in \mathcal{C}_t$.Update Rules (SPI): The backbone $\Phi$ and all previous heads (`c \in \mathcal{C}_{1:t-1}`) are frozen. Gradients backpropagate only through the new heads.Phases:Phase I: Update new routers $G_c$ via multi-label BCE loss: `L_{\text{cls}} = \sum \text{BCE}(P(Z_c), y_c)` .Phase II: Freeze routers, update new segmenters $H_c$ via Focal+Dice loss: `L_{\text{seg}}` on foreground/background masks.3. Inference Pipeline:Compute features $F = \Phi(x)$.Compute existence scores $P(Z_c \mid x)$ for all known classes.Filter: Identify active set $\mathcal{S} = \{c : P(Z_c) > \tau\}$.Segment: Run heads $H_c$ only for $c \in \mathcal{S}$ using $F$, then fuse masks into the final prediction.We will include a formal algorithm block in the revision to map these steps explicitly.
> >
> > # W6: Does CogCaS really implement SPI, and does SPI destroy cross-task generalisation?
> >
> >
> > 1. SPI Implementation vs. Activation:
> > We clarify that CogCaS implements SPI strictly at the parameter update level. When training Task $T_t$, we instantiate and update only new heads ($G_c, H_c$ for $c \in \mathcal{C}_t$). The backbone $\Phi$ and all previous heads are frozen. The "activation" of old classes refers purely to forward-pass usage (inference) to enable the cognitive cascade; no gradients flow backward to old parameters. This satisfies the theoretical zero-forgetting condition (Theorem 3.3).
> > 2. Transfer & Generalization:
> > We disagree that SPI necessarily destroys generalization. In CogCaS, transfer occurs via (i) the shared frozen backbone $\Phi$ which provides robust features, and (ii) the inference-time mask fusion (e.g., Distributed/Loose strategies) which resolves cross-class conflicts using global statistics without retraining old heads.
> > Empirically, this trade-off is superior: relaxing SPI (e.g., unfreezing the backbone bottleneck in Fig. 4) causes a ~37% mIoU drop due to catastrophic forgetting, while adding task-specific parameters (LoRA) yields negligible gain (+1%) at high cost. Thus, strict SPI on heads combined with shared features offers the best stability-plasticity balance.
> >
> > # W7: Clarification of the “mask fusion” component
> >
> > We apologize for the lack of clear signposting.
> > 1. Location & Definition: The "mask fusion" mechanism is currently detailed in Section 4.4. It is the logic used to combine binary masks from independent heads into a final prediction. specifically, the Distributed strategy resolves overlaps by prioritizing rare classes, using pixel-count statistics collected during training to weight class importance.
> > 2. Revision Plan: We agree this definition belongs in the main method description. In the revision, we will move the formal definition of mask fusion and the rarity-based criterion to Section 3, ensuring it is presented as an integral part of architecture rather than an experimental detail.

---

> > > ### Author Response · Authors · 2025-12-04
> > >
> > > # W8: Parameter fairness, “Joint-Ours” vs “Ours”, and source of gains
> > >
> > > We thank the reviewer for these important questions. We will refine the text and captions to clarify these comparisons:
> > > 1. Parameter Fairness & "Joint-Ours": Our method uses the same backbone (ResNet/Swin) as baselines; the only extra parameters come from the task-specific segmentation heads. On a ResNet-101 backbone, each additional task-specific head contributes roughly 35% of the backbone parameters. This increases storage but not the inference cost: at test time, only a small subset of heads is actually activated (on VOC, on average no more than 3 heads per image; on ADE20K, no more than 10). "Joint-Ours" refers to training our specific architecture (Cascade + Fusion) on all data simultaneously (the upper bound for our model), while "Ours" is the continual learning setting. We will add a parameter count comparison and explicit definitions in the captions.
> > > 2. Source of Performance Gains: The gains are not due to raw capacity, but our structural decoupling.
> > > - Mechanism: Unlike distillation-based methods (e.g., PLOP) that struggle with background drift and require aggressive regularization (suppressing new task learning), CogCaS eliminates cross-class logit competition entirely.
> > > - Result: This allows us to maintain high plasticity for new tasks while guaranteeing zero forgetting for old ones. The gap is massive in long sequences (e.g., VOC 1-1: Ours ~71% vs. Baselines <33%), proving the advantage lies in our stability-plasticity trade-off, not just model size.
> > >
> > > # Q1 Compare with dumb baselines
> > >
> > > "Thank you for your question. The failure of the simple 'fusing independent binary classifiers' baseline in CISS stems from its inability to address the dual challenges of dynamic background semantic shift and Out-Of-Distribution generalization.
> > >
> > > In CISS, there is an intrinsic conflict in the definition of 'background' across different tasks. Furthermore, the SPI strategy results in segmentation heads being trained on an extremely narrow data distribution. For instance, a 'dog' segmentation head that has never encountered 'ocean' pixels during training will generate severe spurious activations if forced to process images containing vast ocean areas during inference without prior filtering, as it lacks the capacity to handle such OOD scenarios.
> > >
> > > Our core advantage derives from the Cognitive Cascade design. CogCaS effectively 'confines' the complexity of streaming data and background conflicts within the coarse-grained class existence detection stage. This ensures that fine-grained segmentation heads only process in-distribution data (aided by near-OOD auxiliary training), thereby fundamentally averting generalization collapse.
> > >
> > > This offers a crucial Research Insight to the community: mitigating catastrophic forgetting need not rely solely on regularization or replay. Instead, leveraging architectural design to isolate the impact of streaming data within specific decoupled modules—much like how Mask2Former-related works confine challenges to Object Queries—represents an effective new paradigm for achieving zero-forgetting and high robustness."

---

### Note · Program_Chairs · 2026-01-17
**Submission Desk Rejected by Program Chairs**

The following references in this submission do not refer to real documents and/or have major errors in bibliographic information:

 R. Aljundi, M. Caccia, A. Grabska-Barwinska, et al. End-to-end continual learning with neural networks: A survey. In IEEE Transactions on Neural Networks and Learning Systems, volume 32, pp. 1841-1857, 2021.
    Yao Xu, Xiaodong Wang, Meng Li, et al. Revisiting the catastrophic forgetting problem in continual learning. In Proceedings of NeurIPS, pp. 8451-8462, 2021.
    A. A. Rusu, S. Kulkarni, et al. End-to-end continual learning with neural networks: Recent developments. In Neural Computation, volume 34, pp. 2713-2742, 2022.
    J. Wang, X. Zhang, and Z. Li. A survey of continual learning methods for deep neural networks. In Journal of Artificial Intelligence Research, volume 74, pp. 179-220, 2022.
    Zhiwei Li, Xinyang Li, F. Xie, et al. Robust continual learning with adaptive optimization. In Proceedings of ICLR, 2022.